# Let EEG Models Learn EEG

Yifan Wang [1]  Yijia Ma [2]  Wen Li [2]  Chenyu You [1]

## Abstract

High-fidelity EEG generation is critical for alleviating data scarcity and addressing privacy constraints in large-scale neural modeling. Despite recent progress, most existing approaches formulate EEG generation via discrete denoising objectives, which inadequately reflect the inherently continuous temporal dynamics and spectral structure of neural activity. As a result, these methods often struggle to preserve long-range temporal dependencies and exhibit mismatches in the spectral and temporal structure of the generated signals. In this work, we argue that effective EEG generation requires models that operate directly on the continuous evolution of neural signals. We introduce **Just EEG Transformer (JET)**, a generative framework based on conditional flow matching that models EEG as raw sequences evolving along continuous trajectories. By learning a smooth vector field that transports noise to the EEG data distribution, JET captures temporal continuity and transient dynamics without relying on discretized denoising schemes or domain-specific representations. To ensure that the learned dynamics remain consistent with key properties of EEG signals, we introduce principled constraints that preserve spectral structure, temporal stationarity, and signal-level statistics. Across three large-scale benchmarks, JET consistently achieves state-of-the-art performance, reducing TS-FID by over 40% compared to strong baselines. Extensive analyses show that JET captures key structural properties of neural dynamics, providing a scalable and principled approach to EEG generation. Project page: https://y-research-sbu.github.io/JET/.

## 1. Introduction

Recent years have witnessed rapid progress in generative modeling, reshaping domains ranging from computer vision (Ho et al., 2020; Goodfellow et al., 2014) to natural language processing (Achiam et al., 2023). The field has evolved from adversarial paradigms such as GANs (Goodfellow et al., 2014; Gulrajani et al., 2017) to modern diffusion- and flow-based models (Ho et al., 2020; Song et al., 2021b; Lipman et al., 2022; Liu et al., 2022), achieving unprecedented fidelity and scalability. Architectures have similarly transitioned from convolutional backbones to large Transformers (Peebles & Xie, 2023), enabling high-quality image synthesis (Rombach et al., 2021) and coherent temporal generation. Together, these advances have shifted generative models from auxiliary data augmentation tools to a central mechanism for large-scale data synthesis.

In parallel, large-scale neural modeling has shown increasing promise in decoding brain activity from electroencephalography (EEG). Recent pretrained brain foundation models (Jiang et al., 2024b; Cui et al., 2024; Zhang et al., 2023a; Wang et al., 2024b; 2025; Pan et al., 2024; Liang et al., 2021) demonstrate that data-driven approaches can extract meaningful structure from EEG signals. However, unlike text or images, progress in EEG modeling remains fundamentally constrained by data availability. High-quality clinical recordings are expensive to acquire, require expert annotation, and are subject to strict privacy regulations. Even the largest public datasets, such as the TUH EEG Corpus (Obeid & Picone, 2016), are orders of magnitude smaller than those that underpin modern foundation models. Generative modeling thus presents a natural path forward, yet existing approaches have fallen short of this goal.

When generating raw EEG signals, prevailing generative paradigms (Hartmann et al., 2018; Song et al., 2021a; Puah et al., 2025; Fuhrmeister et al., 2025; Zhang et al., 2021) face inherent difficulties in modeling the structure of neural signals. These methods, which often rely on implicit density estimation or iterative discrete denoising, struggle to navigate the complex, high-dimensional landscape of neural dynamics. These difficulties arise from a fundamental mismatch between common generative formulations and the properties of EEG signals. EEG exhibits power-law spectral structure, non-stationary temporal dynamics, and

[1]Stony Brook University [2]University of Texas Health Center at Houston. Correspondence to: Chenyu You <chenyu.you@stonybrook.edu>.

*Proceedings of the 43rd International Conference on Machine Learning*, Seoul, South Korea. PMLR 306, 2026. Copyright 2026 by the author(s).

heavy-tailed amplitude distributions (Donoghue et al., 2020; Kaplan et al., 2005; Buzsáki & Mizuseki, 2014). Discrete denoising objectives and simple Gaussian priors are ill-suited to these properties, often resulting in spectral bias, temporally static generations, and limited coverage of extreme events. At a high level, this failure stems from the fact that discrete denoising formulations prioritize local reconstruction accuracy under isotropic noise assumptions, whereas EEG generation requires preserving global structure across time, frequency, and amplitude. As a result, small local errors can accumulate over generation steps, resulting in biased spectral content, degraded temporal coherence, and mismatched signal statistics.

This mismatch manifests in three fundamental challenges that are intrinsic to biological time series. (1) *Power-Law Spectral Structure and Spectral Bias*: Neural signals exhibit a power-law spectral density ($1/f^\chi$), with dominant low-frequency oscillations superimposed on a scale-free aperiodic background (Donoghue et al., 2020). In this regime, high-frequency transients carry low energy yet encode dense local information. Objectives that emphasize global energy minimization tend to suppress these components, leading to spectral bias and oversmoothing. (2) *Non-Stationary Temporal Dynamics*: Brain activity is inherently non-stationary, exhibiting continuous, sub-second shifts in energy and connectivity (Kaplan et al., 2005; Khanna et al., 2015). Discrete generation steps often fail to capture such fluid transitions, producing waveforms that appear temporally static or repetitive and lack the evolving structure of real physiological signals. (3) *Heavy-Tailed Amplitude Distributions*: Authentic EEG signals exhibit heavy-tailed amplitude distributions and complex variance patterns (Buzsáki & Mizuseki, 2014; Roberts, 2000). Generative priors based on simple Gaussian assumptions often collapse toward average behavior, failing to cover the diverse amplitude ranges observed in pathological recordings.

In this work, we argue that effective EEG generation requires modeling neural activity as a continuous dynamical process rather than as a sequence of discrete denoising steps. Brain activity evolves smoothly through a high-dimensional state space (Barachant et al., 2011; Gallego et al., 2017; Chaudhuri et al., 2019), suggesting a formulation in which generation follows a continuous trajectory on the neural manifold. This perspective naturally motivates conditional flow matching (Lipman et al., 2022; Li & He, 2025), which learns a vector field that transports a prior distribution to the data distribution, providing a continuous alternative to discretized denoising formulations. As illustrated in Figure 1, this formulation avoids discretized noise removal and better aligns with the continuous evolution of EEG signals.

We introduce **Just EEG Transformer (JET)**, a flow-matching framework designed for high-fidelity EEG gen-

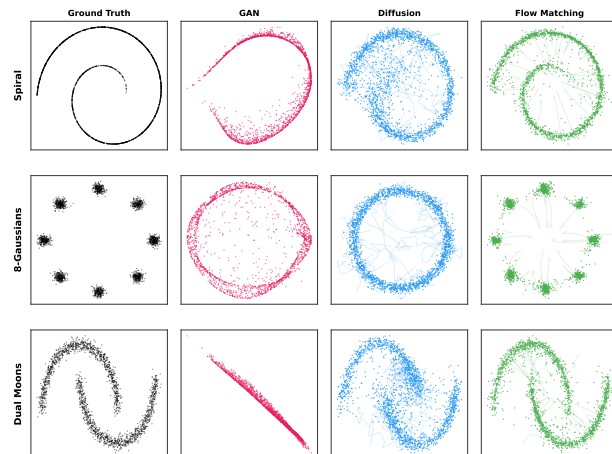

*Figure 1.* **Toy comparison of generative modeling paradigms.** We visualize the learned distributions on three different manifolds. GANs map latent noise directly to data space, diffusion models rely on stochastic denoising dynamics, while flow matching learns a smooth time-dependent vector field whose integral curves realize optimal transport from a dispersed source to the target distribution.

eration. JET models multi-channel EEG as raw sequences evolving along continuous trajectories and leverages the global receptive field of self-attention to capture long-range temporal dependencies and dynamic interactions. To ensure that the learned dynamics remain consistent with key properties of EEG signals, we introduce principled constraints that preserve spectral structure, temporal stationarity, and signal-level statistics. Extensive experiments across three large-scale benchmarks demonstrate that JET achieves state-of-the-art performance, reducing TS-FID by over 40% compared to strong baselines. Further analyses show that JET uncovers intrinsic structure in neural dynamics, providing a scalable and structure-preserving approach to EEG generation. Our contributions are summarized as follows:

- We formulate EEG generation as a continuous dynamical process and introduce **JET**, a flow-matching framework that departs from discrete denoising paradigms.

- We show that modeling raw EEG sequences with Transformers enables effective capture of long-range temporal dependencies and dynamic inter-channel structure.

- We derive a set of principled, structure-preserving constraints that regularize the generative flow to respect spectral, temporal, and statistical properties of EEG.

- We achieve best performance across three large-scale benchmarks, with over 40% reduction in TS-FID, and provide detailed analyses of neural dynamics.

## 2. Related Works

**EEG Models.** The field of cognitive science, especially for brain signals like EEG, has witnessed a paradigm shift towards large scale pretraining, driven by the increasing

availability of extensive datasets such as the TUH EEG Corpus (Obeid & Picone, 2016). Inspired by the success of large models in vision and language domains, a plethora of foundation models have emerged to learn robust EEG representations. BERT-based architectures, including Neuro-BERT (Wu et al., 2022), BrainBERT (Wang et al., 2023), and Brant (Zhang et al., 2023a), utilize masked signal modeling to capture contextual dependencies. More recently, GPT have been adapted for neural signals, as seen in Neuro-GPT (Cui et al., 2024), EEGPT (Wang et al., 2024a), and NeuroLM (Jiang et al., 2024a), demonstrating scalability in sequence modeling. Concurrently, specialized attention mechanisms have been proposed (Jiang et al., 2024b; Chen et al., 2024b; Yang et al., 2023; Chen et al., 2025; Song et al., 2022; Lawhern et al., 2018; Liu et al., 2025a; You et al., 2024; 2025; Wang et al., 2025). Other works have focused on standardizing benchmarks, such as BENDR (Kostas et al., 2021), or refining decoding techniques with CNNs (Schirrmeister et al., 2017) and invariant representations (Song et al., 2023b).

Despite recent progress, most prior work has focused on cross-modal decoding settings, where EEG is treated as a conditional signal to reconstruct external stimuli. A substantial body of prior work has investigated visual stimulus decoding from brain activity, including early approaches such as Brain2Image (Kavasidis et al., 2017) and more recent diffusion-based methods such as DreamDiffusion (Bai et al., 2023), Seeing Beyond the Brain (Chen et al., 2023) and Seeing Through the Brain (Lan et al., 2023). Similarly, a growing body of work explores cross-modal translation from EEG to text, speech, or visual content (Duan et al., 2023; Lee et al., 2023; Zhou et al., 2025; Liu et al., 2025b; Deng et al., 2025; Chen et al., 2024a). While effective for semantic decoding, these approaches primarily treat EEG as a conditioning signal rather than as a generative object itself. In contrast, native generative modeling of EEG signals remains comparatively underexplored. EEG generation is fundamentally challenging due to low signal-to-noise ratios, pronounced non-stationarity, and complex pathological dynamics. As a result, only a limited number of works, such as EEG-GAN (Hartmann et al., 2018) and diffusion-based augmentation methods (Shu et al., 2024), have directly addressed this problem. Recent autoregressive paradigms such as MEG-GPT (Huang et al., 2025) and GPT2MEG (Csaky et al., 2024) have explored related MEG generation, but rely on discretized representations that are inherently misaligned with the continuous nature of neural signals. Given the high cost of data acquisition and the scarcity of clinically meaningful recordings, developing robust EEG generative models is a critical prerequisite for large-scale neural modeling.

**Generative Models.** Generative modeling has experienced a explosion across various domains. GAN (Goodfellow et al., 2014) pioneered this era, evolving through stable training techniques (Gulrajani et al., 2017; Li et al., 2023; You et al., 2019; Ma et al., 2023; Han et al., 2023; Karras et al., 2019; Brock et al., 2019). Subsequently, DDPM (Ho et al., 2020) and Score-Based Generative Models (Song et al., 2021b; Kingma et al., 2021) revolutionized the field by offering stable training and high mode coverage, further refined by improved noise schedules (Nichol & Dhariwal, 2021; Bansal et al., 2023; Daras et al., 2023). To mitigate sampling latency, acceleration techniques (Song et al., 2021a; 2023a; Luo et al., 2023) were introduced. The paradigm further shifted towards latent modeling (van den Oord et al., 2018; Rombach et al., 2021), enabling text-conditional generation at scale (Saharia et al., 2022; Ramesh et al., 2022). Methods (Chang et al., 2022; Han et al., 2024; Wu et al., 2025; Zheng et al., 2025; Sun et al., 2025; 2026; Ren et al., 2025) continue to explore discrete and autoregressive representations. State-of-the-art systems now integrate these advances into massive multimodal simulators (Wang et al., 2026) and controllable frameworks (Zhang et al., 2023b; Shi et al., 2025).

A converging trend in modern generative models is the unification of flow-based methods and Transformer architectures. Flow Matching (Lipman et al., 2022) and Rectified Flow (Liu et al., 2022) provide a simplified objective for training Continuous Normalizing Flows, often yielding straighter trajectories and faster sampling (Liu et al., 2024; Geng et al., 2025; Davis et al., 2024). Other frameworks (Albergo et al., 2023; Karras et al., 2022; Ma et al., 2024) bridge the gap between diffusion and flow matching. Crucially, DiT (Peebles & Xie, 2023) and JiT (Li & He, 2025) demonstrated that standard Vision Transformers can replace U-Nets in diffusion backbones, which validates that minimal inductive bias and strong scalability are sufficient for state-of-the-art generation. Despite the success of simple and scalable generative architectures in vision, their effectiveness for EEG generation remains largely unexplored.

## 3. Method

In this section, we present **Just EEG Transformer (JET)**, a flow-based generative framework for EEG synthesis. JET models EEG generation as a continuous dynamical process by combining conditional flow matching with a Transformer backbone, and incorporates structure-preserving constraints that align the learned dynamics with key statistical properties of EEG signals. Under this formulation, flow matching provides a direct and natural way to model EEG generation as a continuous transformation, avoiding discretized noise schedules that are poorly aligned with neural dynamics. As illustrated in Figure 2, we first introduce the formulation of Conditional Flow Matching in Sec. 3.1. We then describe the JET architecture for modeling raw EEG sequences in Sec. 3.2. Finally, in Sec. 3.3, we present a set of principled

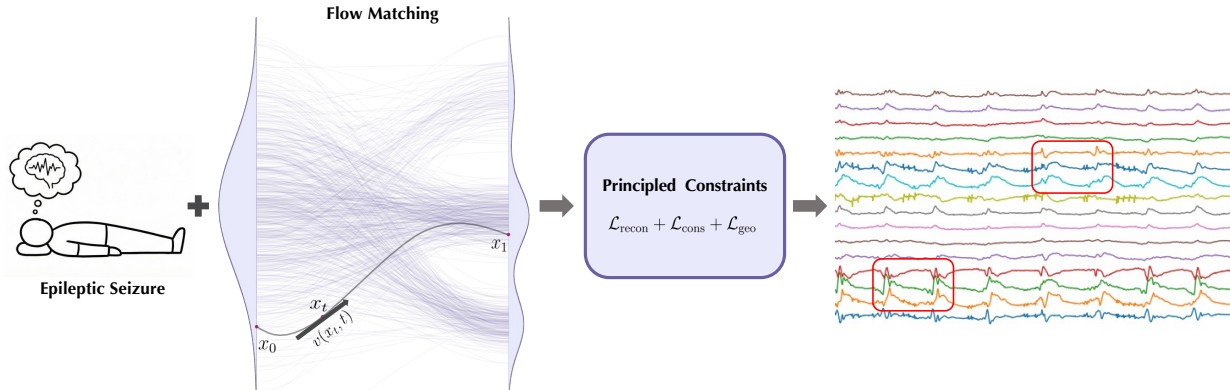

*Figure 2.* **Illustration of JET pipeline.** The model generates multi-channel EEG by learning a continuous vector field $v(x_t, t)$ via flow matching, conditioned on pathological states and regularized by structure-preserving constraints that encode spectral, temporal, and statistical properties of EEG signals.

constraints that regularize the generative flow to preserve spectral, temporal, and statistical structure of EEG signals. Architectural details are provided in Appendix B.1.

### 3.1. Preliminaries

We adopt the Flow Matching framework (Lipman et al., 2022) to model EEG generation as a continuous transformation between a simple prior distribution and the data distribution. Let $\mathbf{x}_1 \sim q(\mathbf{x}_1)$ denote samples from the EEG data distribution, and let $\mathbf{x}_0 \sim p(\mathbf{x}_0) = \mathcal{N}(\mathbf{0}, \mathbf{I})$ denote samples from a standard Gaussian prior. Flow Matching aims to learn a time-dependent vector field $\mathbf{v}_t(\mathbf{x}, t)$ that transports the probability density from $\mathbf{x}_0$ to $\mathbf{x}_1$ as time $t$ evolves from 0 to 1.

To make training tractable, we employ Conditional Flow Matching. Given a pair $(\mathbf{x}_0, \mathbf{x}_1)$, we define a linear interpolation path: $\mathbf{x}_t = t\mathbf{x}_1 + (1 - t)\mathbf{x}_0, t \in [0, 1]$, whose time derivative yields the target vector field $\mathbf{u}_t(\mathbf{x}_t \mid \mathbf{x}_0, \mathbf{x}_1) = \frac{d\mathbf{x}_t}{dt} = \mathbf{x}_1 - \mathbf{x}_0$. The neural network $f_\theta$, instantiated as JET, takes the noisy state $\mathbf{x}_t$, time $t$, and class label $c$ as input, and predicts the vector field $\mathbf{v}_\theta(\mathbf{x}_t, t, c)$. During inference, EEG samples are generated by solving the ordinary differential equation $\frac{d\mathbf{x}_t}{dt} = \mathbf{v}_\theta(\mathbf{x}_t, t, c)$ starting from $\mathbf{x}_0 \sim \mathcal{N}(\mathbf{0}, \mathbf{I})$.

### 3.2. Just EEG Transformer

EEG signals exhibit long-range temporal dependencies and dynamic inter-channel interactions that are not well captured by architectures relying on local receptive fields or fixed graph structures (Kaplan et al., 2005; Khanna et al., 2015; Song et al., 2018). In particular, volume conduction causes activity from a single neural source to propagate simultaneously across multiple electrodes, while functional connectivity patterns evolve continuously over time. These properties violate the locality and stationarity assumptions underlying convolutional architectures and static graph-based models.

JET is designed to model multi-channel EEG directly as raw continuous sequences, avoiding handcrafted feature extraction such as time–frequency transforms or predefined adjacency matrices (Oppenheim & Lim, 2005; Li et al., 2021). By operating on raw EEG signals, the model learns representations that jointly capture spectral and temporal structure in an end-to-end manner, without imposing restrictive inductive biases. Self-attention provides a global receptive field, enabling JET to represent dependencies that span distant time points and spatially distributed channels within a unified sequence representation (Peebles & Xie, 2023; Li & He, 2025). This design allows the model to discover complex neural dynamics, ranging from transient high-frequency events to large-scale functional synchronization, guided solely by the intrinsic structure of the data and the proposed principled constraints.

**Raw EEG Tokenization.** Given a multi-channel EEG segment $\mathbf{X} \in \mathbb{R}^{C \times T}$, where $C$ denotes the number of channels and $T$ the number of time points, we adopt a patch-based tokenization strategy inspired by previous works (Wang et al., 2024b; Klein et al., 2025). Specifically, the temporal axis is divided into $N = T/P$ non-overlapping patches of length $P$, resulting in a reshaped representation $\mathbf{X}_p \in \mathbb{R}^{C \times N \times P}$. Each temporal patch $\mathbf{p} \in \mathbb{R}^P$ is then linearly projected into a $D$-dimensional embedding. Crucially, unlike standard Vision Transformers that flatten spatial dimensions prior to projection, we preserve channel identity during the initial embedding stage to maintain spatial structure. This yields an input token sequence $\mathbf{Z}_0 \in \mathbb{R}^{(C \cdot N) \times D}$. Learnable positional embeddings are added to encode temporal order.

**Transformer Backbone.** JET consists of a stack of standard Transformer blocks with multi-head self-attention and feed-forward layers. Following DiT (Peebles & Xie, 2023) and JiT (Li & He, 2025), conditioning information, including diffusion time $t$ and class label $c$, is injected via adaptive layer normalization. Specifically, the scale and shift parameters of each normalization layer are predicted from the sum of the time embedding and class embedding, ensuring that

the generative process remains fully controllable throughout the flow trajectory. This design allows the model to jointly reason over temporal evolution and inter-channel interactions within a unified representation.

**Adaptive Class-Balanced Sampling.** Real-world clinical EEG datasets inherently exhibit severe class imbalance, where normal background activity vastly outnumbers rare but clinically significant pathological events such as seizures (Obeid & Picone, 2016). Under uniform sampling, generative models tend to bias toward majority classes, which can lead to mode collapse and poor coverage of pathological signal patterns. To mitigate this issue, we employ an adaptive class-balanced sampling strategy. Let $N_c$ denote the number of samples for class $c \in \mathcal{C}$. Each training sample $x_i$ from class $c$ is assigned a sampling probability: $p_i \propto \frac{1}{N_c^\alpha}$, where $\alpha \in [0,1]$ controls the strength of re-balancing. This strategy effectively constructs a training distribution with approximately uniform expected class frequency, encouraging the model to learn robust representations of under-represented pathological patterns without overfitting to dominant normal rhythms.

### 3.3. Principled Constraints

Standard flow matching objectives rely on Euclidean regression losses under Gaussian noise assumptions. While effective for many modalities, such objectives are insufficient for EEG signals, which exhibit heavy-tailed amplitudes, non-stationary dynamics, and structured spectral content. As discussed in Sec. 1, this mismatch often leads to spectral bias and degraded temporal coherence when applying generic objectives to EEG generation.

Here, *principled* refers to constraints that arise naturally from the modeling assumptions and the statistical structure of EEG signals, rather than ad hoc regularization. These constraints serve to regularize the learned flow so that it preserves key properties of EEG data along reconstruction, statistical, and spatiotemporal dimensions. Together, these constraints act as a structural regularizer of the generative flow. We introduce three complementary constraints described below. More theoretical motivation are in Appendix C.

**Reconstruction with Laplacian Priors.** EEG recordings frequently contain impulsive, non-Gaussian artifacts arising from muscle activity, electrode movement, or external noise. An $L_2$ reconstruction loss corresponds to a Gaussian likelihood and is therefore sensitive to such outliers. We instead adopt an $L_1$ reconstruction loss, which implicitly assumes a Laplacian prior and promotes robustness to transient deviations while preserving sharp neural events:

$$\hat{\mathbf{x}}_1 = \mathbf{x}_t + (1-t)\, v_\theta(\mathbf{x}_t, t, c), \; \mathcal{L}_{\text{recon}} = \mathbb{E}_t \left\| \mathbf{x}_1 - \hat{\mathbf{x}}_1 \right\|_1. \quad (1)$$

**Statistical Consistency.** To prevent drift in amplitude statistics, we constrain the first- and second-order moments of generated signals to match those of real EEG data. Concretely, $\mu(\cdot)$ and $\sigma(\cdot)$ denote the per-channel temporal mean and standard deviation, computed along the time axis $T$ and then averaged over channels and batch. This encourages the generative flow to remain on the same statistical manifold as the data distribution, stabilizing signal amplitudes across time and channels:

$$\mathcal{L}_{\text{cons}} = \lambda_{\text{cons}}(\|\mu(\mathbf{x}_1) - \mu(\hat{\mathbf{x}}_1)\|_1 + \|\sigma(\mathbf{x}_1) - \sigma(\hat{\mathbf{x}}_1)\|_1). \quad (2)$$

**Spatiotemporal Structure.** EEG signals exhibit structured temporal evolution and band-limited spectral content. To preserve spectral smoothness, we apply a temporal total variation regularizer, which suppresses spurious high-frequency fluctuations while retaining physiologically meaningful dynamics. In addition, we encourage structural alignment between generated and real signals by maximizing the Pearson correlation, which is invariant to amplitude scaling and captures waveform morphology:

$$\begin{aligned} \mathcal{L}_{\text{geo}} &= \lambda_{\text{tv}} \mathcal{L}_{\text{tv}} + \lambda_{\text{corr}} \mathcal{L}_{\text{corr}} \\ &= \lambda_{\text{tv}} \frac{1}{T} \sum_t \|\nabla_t \hat{\mathbf{x}}_1\|_1 + \lambda_{\text{corr}} \left(1 - \rho(\mathbf{x}_1, \hat{\mathbf{x}}_1)\right). \end{aligned} \quad (3)$$

**Total Objective.** The final training objective combines the flow matching loss with the proposed constraints:

$$\mathcal{L}_{\text{total}} = \mathcal{L}_{\text{recon}} + \mathcal{L}_{\text{cons}} + \mathcal{L}_{\text{geo}}. \quad (4)$$

## 4. Experiments

In this section, we conduct comprehensive experiments to validate the performance of JET across multiple dimensions. We first detail the large-scale benchmarks and our multi-dimensional evaluation protocol in Section 4.1. Next, we demonstrate JET's superior quantitative performance and downstream utility compared to leading baselines in Section 4.2. We then conduct a fine-grained physiological analysis to verify the preservation of key dynamical structure in Section 4.3. Finally, we provide ablation studies to isolate the contributions of our noise strategy and principled constraints in Section 4.4.

### 4.1. Dataset and Metrics

**Datasets.** We evaluate JET on three large-scale datasets of the TUH Corpus (Obeid & Picone, 2016): TUAB (Abnormal), TUEV (Events) and TUSZ (Seizures). Collectively, these benchmarks comprise over 10,000 clinical sessions, representing the largest open-source repository for neurological time-series. This diverse suite rigorously tests the model across three critical regimes: global pathological distributions (TUAB), fine-grained transient waveforms (TUEV), and extreme non-stationary seizure dynamics (TUSZ). For

| Method | TUAB | | | TUEV | | | TUSZ | | |
|---|---|---|---|---|---|---|---|---|---|
| | TS-FID ↓ | Sil. ↑ | Δ Acc ↑ | TS-FID ↓ | Sil. ↑ | Δ Acc ↑ | TS-FID ↓ | Sil. ↑ | Δ Acc ↑ |
| EEG-GAN | 324.18 | 0.786 | +0.000 | 448.65 | 0.667 | -0.004 | 274.37 | 0.891 | +0.001 |
| Vanilla Diffusion | 342.91 | 0.710 | -0.002 | 415.82 | 0.703 | -0.000 | 300.47 | 0.746 | +0.000 |
| **Ours** | **188.27** | **0.995** | **+0.029** | **235.86** | **0.983** | **+0.032** | **151.27** | **0.987** | **+0.017** |

*Table 1.* **Quantitative comparison on TUAB, TUEV, and TUSZ datasets.** We evaluate generation quality (TS-FID ↓), conditional consistency (Sil. ↑), and downstream utility (Δ Acc ↑). **Bold** indicates the best performance.

preprocessing, we align all recordings to the standard channel setting and sample rate to ensure spectral consistency. Detailed dataset statistics, version specifications, and filtering protocols are provided in Appendix A.1.

**Evaluation Metrics.** To provide a holistic assessment of generative quality, we adhere to a multi-dimensional evaluation protocol covering distributional fidelity, structural diversity, and clinical utility: (1) **Time-Series FID (TS-FID):** Since standard FID is not designed for physiological time series (detailed in Appendix A.3), we adopt a domain-specific TS-FID. This metric computes the Fréchet distance between real and generated signals in a spectral feature space, quantifying alignment of spectral structure between the two distributions. (2) **Silhouette Score (Sil.):** To verify conditional consistency and rule out mode collapse, we calculate the Silhouette Score (Teng et al., 2025) on the generated embeddings. This quantifies how well synthesized EEG cluster according to their pathological labels. (3) **Downstream Utility (Δ Acc):** We measure practical utility by using synthetic data to augment a state-of-the-art EEG classifier CbraMod (Wang et al., 2024b). A positive accuracy gain indicates that the generated samples possess high-fidelity discriminative features that aid the decision boundary. Mathematical formulations and implementation details for all metrics are provided in Appendix A.2.

### 4.2. Quantitative Comparison

We compare our proposed framework against two representative generative baselines: EEG-GAN (Hartmann et al., 2018), a GAN-based approach adapted for EEG synthesis, and Vanilla Diffusion (Song et al., 2021a), a standard diffusion probabilistic model without our specific architectural enhancements. Table 1 summarizes the performance across all three datasets. For implementation and computation detail, please refer to Appendix B.2 and Appendix B.3

**Generation Quality.** As demonstrated in Table 1, our method achieves state-of-the-art performance across all metrics and datasets, outperforming baselines by a substantial margin. In terms of distributional fidelity, we observe a reduction in TS-FID scores. This indicates that our architectural design effectively captures the complex spectral-temporal dynamics of EEG signals that standard diffusion models and GANs fail to model accurately.

**Label Consistency.** The Silhouette Scores further validate the semantic correctness of our generation. While baselines hover between 0.66 and 0.89, our method consistently achieves near-perfect scores across all datasets. This suggests that our model maintains strict adherence to conditioning labels, generating samples that form distinct, well-separated clusters in the feature space without suffering from mode collapse or class confusion.

**Downstream Performance.** Most notably, the Δ Acc metric highlights the practical value of our method as a data augmentation tool. Baselines exhibit negligible or even negative improvements, implying that their generated samples introduce noise or distributional shifts that hinder classifier training. In contrast, our method consistently yields positive accuracy gains, improving downstream performance. This confirms that our synthetic data is not only realistic in appearance but also distributionally aligned with the ground truth, effectively supplementing the training distribution for discriminative tasks.

**Computational Efficiency.** Beyond fidelity, JET also offers computational advantages over discrete denoising schemes and token-by-token decoding. Generating a batch of 32 ten-second samples takes $4.78$ s for JET, compared with $7.01$ s for diffusion. EEG-GAN is marginally faster at $4.12$ s, but yields substantially lower fidelity and poor downstream utility. JET therefore achieves the best fidelity–latency trade-off. Full efficiency statistics are in Appendix B.3.

### 4.3. Physiological Analysis

To go beyond aggregate metrics and assess whether JET preserves key physiological structure, we conduct a fine-grained analysis grounded in established properties of EEG signals. We structure this analysis along three fundamental dimensions identified in Sec. 1: spectral structure, temporal dynamics, and statistical distributions. Together, these analyses examine whether the proposed constraints effectively address the limitations observed in prior generative paradigms. Additional analysis results are in Appendix D.2.

▷ **Can JET Overcome Spectral Bias under Power-Law Scaling?** We first examine whether JET preserves the power-law spectral structure ($1/f^\chi$) and low-energy high-frequency components of EEG signals (Donoghue et al.,

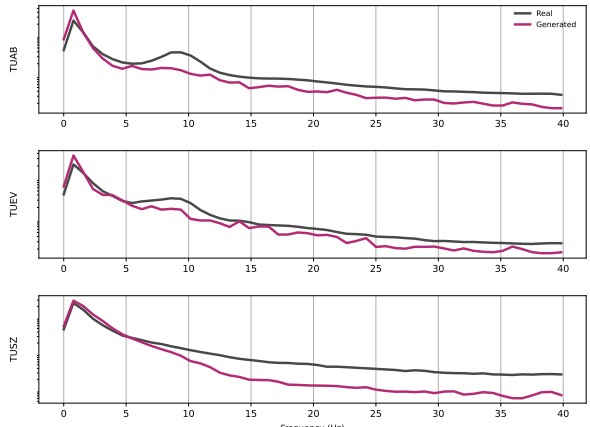

*Figure 3.* **Evaluation of spectral structure.** JET accurately captures the $1/f^\chi$ spectral scaling and prominent $\alpha$-band peaks, mitigating the spectral bias that suppresses high-frequency components in prior methods.

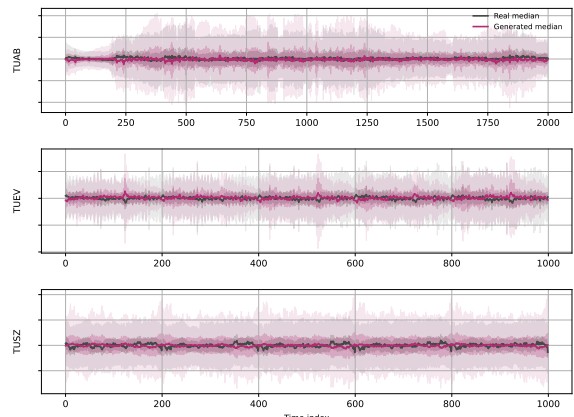

*Figure 4.* **Evaluation of temporal dynamics.** Temporal evolution of signal amplitude distributions. JET captures non-stationary energy fluctuations over time while maintaining stable amplitude statistics, avoiding temporal drift and variance collapse.

| Method | $W_1(\text{slope}) \downarrow$ | $W_1(D_\mu) \downarrow$ | $W_1(D_\sigma) \downarrow$ |
|---|---|---|---|
| Real split | 0.008 | 0.012 | 0.010 |
| EEG-GAN | 0.065 | 0.078 | 0.071 |
| Vanilla Diff. | 0.051 | 0.063 | 0.058 |
| **JET** | **0.015** | **0.021** | **0.018** |

*Table 2.* **Spurious drift analysis on TUEV.** JET's per-segment drift statistics stay close to the real-vs-real floor.

2020). Conventional objectives often suppress these components due to spectral bias, leading to oversmoothed generations. Figure 3 compares the power spectral density (PSD) of generated and real signals, revealing strong alignment across frequency bands:

(1) **Low-Frequency Precision ($\delta$-band):** In the $0 - 5$ Hz range, which contains high-amplitude pathological slow waves and fundamental background rhythms, the generated spectra closely follow the ground truth across all datasets. This alignment indicates that JET preserves dominant low-frequency components without introducing baseline distortion or amplitude drift, a common failure mode in long-sequence generation. Accurate modeling in this band is particularly important for pathological EEG, where slow-wave activity often carries critical clinical information.

(2) **Structural Preservation in Mid-Frequencies:** In the $\alpha$-band ($8 - 13$ Hz), especially in TUAB and TUEV, JET reproduces distinct $\alpha$-band peaks rather than collapsing to a smooth $1/f$ profile. This demonstrates that the model captures structured oscillatory activity beyond global spectral trends. Such behavior suggests that JET learns specific rhythmic signatures associated with functional brain states, rather than merely matching marginal spectral statistics.

(3) **High-Frequency Selectivity:** For frequencies above 15 Hz ($\beta/\gamma$ bands), generated spectra exhibit mild attenuation relative to the ground truth. Rather than indicating spectral collapse, this pattern reflects selective suppression of unstructured high-frequency components while retaining coherent neural activity. Given that high-frequency EEG recordings are often contaminated by electromyogenic and sensor noise, this behavior suggests that JET prioritizes physiologically meaningful dynamics over stochastic artifacts present in raw signals.

▷ **Can JET Model Non-Stationary Temporal Dynamics?** Next, we examine whether JET captures non-stationary temporal dynamics (Kaplan et al., 2005; Khanna

et al., 2015) while avoiding pathological drift over long sequences. Figure 4 shows the temporal evolution of signal envelopes, indicating that the generated signals maintain stable amplitude statistics over time without baseline drift or variance explosion.

(1) **Baseline Stability:** The median of the generated signals remains centered around the real signals throughout the entire time course across all datasets. This indicates that the model successfully prevents baseline drift, a common failure mode in long-sequence generation.

(2) **Consistent Variance Structure:** The inter-percentile bands (shaded regions) remain aligned with the real bands within the window. Unlike baselines that suffer from error accumulation leading to variance explosion or collapse, our flow-based approach preserves the signal's energy profile consistently over time.

(3) **Envelope Alignment:** The generated variance envelope closely tracks the ground truth. Notably in TUEV, where the real data exhibits bursty high-amplitude transients, the generated distribution's outer quantiles effectively cover these regions, demonstrating that the model accommodates non-stationary temporal events while maintaining bounded and stable signal statistics.

To quantify spurious drift directly, we compute per-segment statistics from the channel-aggregated RMS envelope $e_i(t) = \sqrt{\frac{1}{C}\sum_c x_{i,c,t}^2}$, namely the linear slope of $e_i(t)$ together with the moment-drift scores $D_\mu = \frac{1}{C}\sum_c |\mu_c^{\text{first}} -$

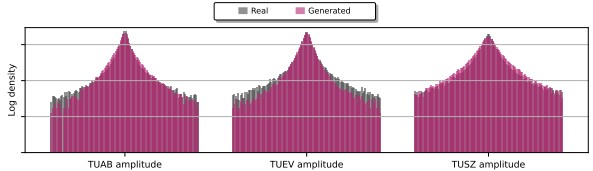

*Figure 5.* **Analysis of amplitude distributions.** Log-density histograms confirm that JET accurately reconstructs the non-Gaussian, heavy-tailed distributions of raw EEG, effectively covering the diverse amplitude ranges of pathological recordings.

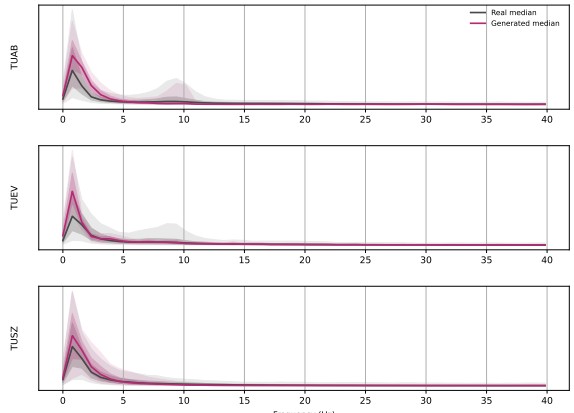

*Figure 6.* **Analysis of population diversity.** The extensive overlap in spectral variance envelopes confirms that JET captures the complex variances of the population, avoiding the mode collapse typical of Gaussian-based priors.

$\mu_c^{\text{last}}|$ and $D_\sigma = \frac{1}{C}\sum_c |\sigma_c^{\text{first}} - \sigma_c^{\text{last}}|$ between the first and last quarter of each segment. We then compare the distribution of each statistic against held-out real data via the 1-Wasserstein distance $W_1$, using a real-vs-real split as the empirical floor (Table 2). JET's drift distances stay within $2\times$ of the real-vs-real floor, while EEG-GAN and Vanilla Diffusion are substantially farther.

▷ **Can JET Align Heavy-Tailed Distributions?** Finally, we investigate the alignment with the heavy-tailed, non-Gaussian distributions typical of pathological populations (Buzsáki & Mizuseki, 2014; Roberts, 2000), ensuring the model avoids mode collapse. We analyze marginal amplitude density via Figure 5 and population-level spectral stability using Figure 6:

(1) **Heavy-Tail Reconstruction:** Within the valid signal window shown in Figure 5, the generated log-density exhibits strong alignment with the ground truth. The model accurately reproduces both the sharp central peak and the heavy-tailed decay of the amplitude distribution, demonstrating accurate modeling of EEG signal statistics without being dominated by sparse outlier noise.

(2) **Avoidance of Mode Collapse:** In Figure 6, the substantial overlap between the generated and real confidence intervals confirms that our model avoids mode collapse. The generated samples exhibit a wide dispersion in spectral

| Noise Strategy | TS-FID ↓ | | |
|---|---|---|---|
| | **TUAB** | **TUEV** | **TUSZ** |
| Zero | 1615.32 | 1964.93 | 1958.24 |
| **Gaussian** | **188.27** | **235.86** | **151.27** |

*Table 3.* Ablation study on noise space initialization. **Bold** indicates best performance.

| Constraints Configuration | | | | TS-FID ↓ | | |
|---|---|---|---|---|---|---|
| $\mathcal{L}_{\text{recon}}$ | $\mathcal{L}_{\text{cons}}$ | $\mathcal{L}_{\text{tv}}$ | $\mathcal{L}_{\text{corr}}$ | **TUAB** | **TUEV** | **TUSZ** |
| ✓ | | | | 231.19 | 287.81 | 221.74 |
| ✓ | ✓ | | | 228.87 | 281.70 | 209.99 |
| ✓ | | ✓ | | 219.45 | 266.61 | 210.00 |
| ✓ | | | ✓ | 221.26 | 278.01 | 200.87 |
| ✓ | ✓ | ✓ | | 211.17 | 250.05 | 197.29 |
| ✓ | ✓ | | ✓ | 197.09 | 239.42 | 178.66 |
| ✓ | | ✓ | ✓ | 190.28 | 241.40 | 164.39 |
| ✓ | ✓ | ✓ | ✓ | **188.27** | **235.86** | **151.27** |

*Table 4.* Ablation study on principled constraints components. ✓ indicates inclusion. **Bold** indicates best performance.

power, mirroring the inter-subject and inter-state variability found in the ground truth, rather than converging to a single deterministic profile.

(3) **Frequency-Dependent Stochasticity:** Demonstrated in Figure 6, the model correctly learns the natural stochasticity of slow-wave activities ($< 5$ Hz), matching the real confidence intervals with high precision. Conversely, in higher frequencies, the generated distribution exhibits a slightly narrower spread. This reflects a conservative constraint that prioritizes robust neural features over sporadic, high-variance artifacts or noisy recording conditions.

### 4.4. Ablation Study

To validate the theoretical foundations of JET, we dissect two design choices in this ablation section: the topological requirements of the noise space and the necessity of principled constraints. We evaluate these components using TS-FID across all benchmarks. For more ablation results, please refer to Appendix D.1.

▷ **Gaussian Noise vs. Zero Noise.** A natural concern in EEG generation is whether high-variance Gaussian noise may overwhelm subtle neural structure during training. To examine this, we compare the standard Gaussian base distribution $\mathcal{N}(\mathbf{0}, \mathbf{I})$ with a degenerate zero-noise initialization $\delta(\mathbf{0})$. As shown in Table 3, initializing from zero leads to catastrophic performance degradation across all datasets. This indicates that a non-degenerate base distribution is necessary for learning a transport map that covers the highly multimodal EEG data distribution. We attribute this behavior to fundamental topological constraints:

(1) **Zero Noise Leads to Mode Collapse.** Initializing from a single point forces the model to expand a degenerate distribution into a complex, multimodal data distribution, resulting in an ill-posed one-to-many mapping that cannot be represented by a continuous flow.

| Configuration | TS-FID↓ | PSD Slope↓ | Temp. Env.↓ | Hjorth Act.↓ | Hjorth Mob.↓ | Hjorth Comp.↓ |
|---|---|---|---|---|---|---|
| $\mathcal{L}_{\mathrm{recon}}$ only | 287.81 | 0.37 | 0.38 | 0.30 | 0.24 | 0.33 |
| $+\mathcal{L}_{\mathrm{cons}}$ | 281.70 | 0.34 | 0.27 | 0.19 | 0.21 | 0.29 |
| $+\mathcal{L}_{\mathrm{tv}}$ | 266.61 | 0.23 | 0.33 | 0.27 | 0.15 | 0.21 |
| $+\mathcal{L}_{\mathrm{corr}}$ | 278.01 | 0.32 | 0.25 | 0.26 | 0.19 | 0.27 |
| **Full** | **235.86** | **0.18** | **0.18** | **0.14** | **0.11** | **0.15** |

*Table 5.* **Constraint-wise effect on signal-level diagnostics (TUEV).** None of the six diagnostic metrics are directly optimized in training; each constraint acts on a complementary failure mode.

**(2) Gaussian Noise Enables Broad Coverage.** In contrast, a Gaussian base distribution provides full support over the ambient space, enabling the learned vector field to smoothly transport probability mass onto the target EEG distribution. This non-degenerate initialization is essential for effective coverage of complex neural dynamics under flow-based generative modeling.

▷ **Principled Constraints.** Standard generative models typically rely on pixel-wise reconstruction constraint ($\mathcal{L}_{\mathrm{recon}}$). We posit that for time-series with complex spatiotemporal dependencies, Euclidean distance alone is insufficient. We evaluate how explicitly constraining conservation ($\mathcal{L}_{\mathrm{cons}}$), smoothness ($\mathcal{L}_{\mathrm{tv}}$), and correlation ($\mathcal{L}_{\mathrm{corr}}$) improves preservation of spectral, temporal, and statistical structure in generated EEG signals. Table 4 and Table 5 demonstrates the hierarchy of importance and the joint effect among the loss terms:

(1) $\mathcal{L}_{\mathbf{recon}}$. The configuration relying solely on $\mathcal{L}_{\mathrm{recon}}$ yields the poorest performance across all benchmarks. This confirms that optimizing for pixel-wise distance alone often results in mean-seeking behavior, producing blurred predictions that lack high-frequency texture.

(2) $\mathcal{L}_{\mathbf{tv}}$ **vs** $\mathcal{L}_{\mathbf{corr}}$. Among single-term additions, structural constraints ($\mathcal{L}_{\mathrm{tv}}$ and $\mathcal{L}_{\mathrm{corr}}$) provide more significant gains. specifically, $\mathcal{L}_{\mathrm{tv}}$ drastically reduces high-frequency noise, while $\mathcal{L}_{\mathrm{corr}}$ ensures phase alignment.

(3) $\mathcal{L}_{\mathbf{cons}}$. It acts as a necessary regularizer for the energy envelope, preventing the physical implausibility that arise when optimizing for smoothness or correlation in isolation.

▷ **Patch Size.** Because patch-based tokenization introduces a temporal-granularity inductive bias, we ablate the patch size $P$ on TUEV in Table 6, reporting both distributional fidelity (TS-FID) and event-level morphology metrics (Pearson correlation and DTW on event-centered windows). Overall distributional fidelity is stable for $P \in \{50, 100, 200\}$, while finer patches preserve transient event morphology slightly better at the cost of longer token sequences for self-attention ($2\times$ for $P{=}100$, $4\times$ for $P{=}50$, relative to the default $P{=}200$). $P{=}400$ degrades both global and local quality. We therefore adopt $P{=}200$ as an efficiency–fidelity compromise.

▷ **Constraint Design vs. Tokenizer-Style Losses.** Sev-

| Patch Size | TS-FID↓ | Event Corr.↑ | Event DTW↓ |
|---|---|---|---|
| 50 | 237.12 | 0.72 | 8.1 |
| 100 | 233.32 | 0.71 | 8.3 |
| **200 (default)** | 235.86 | 0.68 | 8.8 |
| 400 | 241.23 | 0.61 | 10.2 |

*Table 6.* **Patch size ablation.** Distributional fidelity is stable across $P$; finer patches marginally improve event morphology at the cost of longer token sequences.

| Objective Design | TUAB↓ | TUEV↓ | TUSZ↓ |
|---|---|---|---|
| BrainOmni-style tokenizer losses | 314.87 | 471.15 | 369.28 |
| **JET principled constraints** | **188.27** | **235.86** | **151.27** |

*Table 7.* **Constraint design comparison (TS-FID).** Replacing JET's principled constraints with a tokenizer-style loss set inside the same backbone degrades fidelity across all three benchmarks.

eral individual terms such as $L_1$ reconstruction or Pearson-correlation alignment have been used in prior neural-signal modeling like the tokenizer losses in BrainOmni (Xiao et al., 2025). Our claim is not that any single term is unprecedented, but that the *combination* is structurally aligned with raw EEG. To isolate this, we replace our constraints with a BrainOmni-style tokenizer loss set inside the JET architecture and re-train on all three TUH benchmarks (Table 7). The EEG-specific principled constraints achieve substantially lower TS-FID than the tokenizer-style baseline, suggesting that the gain stems from constraint design rather than from merely adding more loss terms.

## 5. Conclusion

In this work, we introduced **Just EEG Transformer (JET)**, a generative framework that formulates EEG synthesis as a continuous dynamical process based on flow matching and principled constraints. Through extensive experiments, we show that modeling EEG generation in this manner effectively addresses key limitations of existing denoising-based approaches. JET achieves state-of-the-art fidelity while consistently capturing key statistical and dynamical properties of EEG signals, including power-law spectral structure, non-stationary temporal dynamics, and heavy-tailed amplitude distributions. By providing a scalable and principled approach to high-fidelity EEG generation, this work contributes a foundation for advancing data-driven neural modeling under realistic data constraints.

## Impact Statement

This paper presents work whose goal is to advance the field of neuroscience, cognitive science, and machine learning. There are many potential societal consequences of our work, none of which we feel must be specifically highlighted here.

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

# A. Dataset and Evaluation Metrics

In this section, we will introduce the details of datasets and evaluation metrics used in this work.

## A.1. Dataset Details and Preprocessing

**Dataset Overview and Specifications.**   We conduct our evaluation on the Temple University Hospital EEG Corpus (TUH EEG) (Obeid & Picone, 2016), the largest open-source repository of clinical EEG recordings. To ensure a comprehensive assessment of JET's generative capabilities across diverse neurological regimes, we utilize three distinct subsets: the Abnormal EEG Corpus (TUAB), the EEG Events Corpus (TUEV), and the Seizure Corpus (TUSZ). All data are stored as preprocessed `float32` pickled arrays, arranged as fixed 16-channel recordings.

**TUAB (Abnormal Corpus).**   This dataset focuses on the binary classification of routine clinical recordings as either *normal* or *abnormal*. The abnormal class encompasses a wide range of pathologies, providing a testbed for modeling general neurological deviations. We utilize the standard train-eval split, where samples are reshaped to a temporal resolution of $16 \times 2000$ (corresponding to 10 seconds at 200 Hz).

**TUEV (Events Corpus).**   To evaluate the model's ability to capture specific fine-grained morphological features, we employ the TUEV dataset, which annotates specific artifactual and epileptiform events. The labels are mapped to six distinct classes with indices 0–5: Spike and Wave (spsw), Generalized Periodic Epileptiform Discharges (gped), Periodic Lateralized Epileptiform Discharges (pled), Eye Movements (eyem), Artifacts (artf), and Background (bckg). Samples in this subset are processed with a window size of $16 \times 1000$, focusing on capturing the transient nature of these short-duration events.

**TUSZ (Seizure Corpus).**   As the gold standard for seizure detection, TUSZ provides a rigorous benchmark for synthesizing high-amplitude, non-stationary seizure activities. The dataset is organized into binary *seizure* and *background* segments. Similar to TUEV, we utilize a segment length of 1000 time points ($16 \times 1000$) to align with the rapid evolution of ictal states.

**Preprocessing Pipeline.**   We employ a uniform preprocessing pipeline across all datasets to ensure consistency. First, strictly 16 standard clinical channels are selected to form the input tensor. Second, to ensure numerical stability within the diffusion process and flow matching vector field, we apply a global normalization strategy where raw signal amplitudes are scaled by a factor of 0.01 (1/100) at load time. Table 8 summarizes the specific configurations, sample dimensions, and class distributions for each subset.

| Dataset | Task Type | Classes | Input Shape | Scale | Description |
|---------|-----------|---------|-------------|-------|-------------|
| **TUAB** | Binary Classification | 2 | $16 \times 2000$ | 0.01 | Normal vs. Abnormal Pathology |
| **TUEV** | Event Recognition | 6 | $16 \times 1000$ | 0.01 | SPSW, GPED, PLED, EYEM, ARTF, BCKG |
| **TUSZ** | Seizure Detection | 2 | $16 \times 1000$ | 0.01 | Seizure vs. Background |

*Table 8.* **Summary of dataset specifications.** We report the input dimensions ($C \times T$), the number of distinct classes, and the semantic description of the tasks for TUAB, TUEV, and TUSZ.

## A.2. Evaluation Metrics Details

**Time-Series Fréchet Inception Distance (TS-FID).**   To extend Fréchet Inception Distance (FID) to neural time-series, we construct a task-specific feature extractor $\phi(\cdot)$ that maps a raw EEG sample $\mathbf{x} \in \mathbb{R}^{C \times T}$ to a fixed-dimensional representation $\mathbf{f} \in \mathbb{R}^D$. The extractor operates in the frequency domain, using Fourier magnitude representations to emphasize spectral structure while remaining invariant to phase. Optional high-frequency truncation is applied to suppress measurement noise. The resulting spectra are then aggregated through spectral and spatial pooling to produce a compact feature vector suitable for distributional comparison.

$$\mathbf{f} = \text{Norm}\left(\log\left(1 + \text{Pool}_{\text{spatial}}\left(\text{Pool}_{\text{spectral}}\left(|\mathcal{F}(\mathbf{x})|_{0:f_{cut}}\right)\right)\right)\right) \tag{5}$$

where $\mathcal{F}(\cdot)$ denotes the FFT. We pool the spectral magnitude into $K_{feat} = 256$ frequency bins (default in evaluation) and $K_{spat} = 4$ spatial regions, followed by log1p and per-sample standardization. The TS-FID is then calculated as:

$$\text{TS-FID} = \|\mu_r - \mu_g\|_2^2 + \text{Tr}(\Sigma_r + \Sigma_g - 2(\Sigma_r \Sigma_g)^{1/2}) \tag{6}$$

where $(\mu_r, \Sigma_r)$ and $(\mu_g, \Sigma_g)$ are the statistics of the real and generated distributions, respectively.

**Silhouette Score.** We assess conditional consistency and mode separation using a clustering-based silhouette score. Each EEG sample is first converted into a low-dimensional feature representation by computing bandpower statistics. Specifically, we estimate the power spectral density (PSD) using Welch's method and average power within the standard frequency bands $\{\delta, \theta, \alpha, \beta, \gamma\}$ corresponding to $0.5 – 4, 4 – 8, 8 – 13, 13 – 30$, and $30 – 45$ Hz. This yields a compact feature vector that captures coarse spectral characteristics of EEG signals. To evaluate cluster structure jointly, we concatenate real and generated features and apply KMeans clustering with $k$. The silhouette coefficient for each sample $i$ is then computed by comparing its mean intra-cluster distance (cohesion $a(i)$) to the mean distance to the nearest neighboring cluster (separation $b(i)$):

$$s(i) = \frac{b(i) - a(i)}{\max\{a(i), b(i)\}}. \tag{7}$$

Here we report the mean silhouette score across all samples. Higher values indicate more compact clusters with clearer separation, reflecting stronger alignment between generated samples and their intended conditioning.

**Downstream Utility ($\Delta$ Acc).** We evaluate the effectiveness of synthetic data for augmentation by measuring the change in classification accuracy on a held-out test set.

$$\Delta\text{Acc} = \text{Acc}(\mathcal{D}_{\text{train}} \cup \mathcal{D}_{\text{gen}}) - \text{Acc}(\mathcal{D}_{\text{train}}). \tag{8}$$

We use CbraMod (Wang et al., 2024b) as the evaluation backbone, which has been widely adopted in recent EEG studies.

### A.3. Rationale for Domain-Specific Evaluation Metrics

In this work, we prioritize the domain-specific TS-FID over generic image-based or raw time-domain metrics. This choice is driven by two fundamental limitations of standard generative evaluation protocols when applied to physiological time-series.

▷ **Incompatibility of ImageNet embeddings.** A common practice in spectrogram generation is to compute FID using an Inception network pretrained on ImageNet. However, this approach is fundamentally flawed for EEG spectral analysis due to the lack of translation invariance in the frequency domain. In natural images, an object retains its semantic meaning regardless of its spatial position $(x, y)$. In contrast, the axes of a spectrogram represent Time and Frequency. A vertical translation in a spectrogram fundamentally alters the biological meaning of the signal (e.g., shifting an Alpha rhythm at 10Hz to a Gamma rhythm at 40Hz changes the neural state from relaxation to active processing). Since Inception features are designed to be translation-invariant, they are incapable of distinguishing these critical spectral shifts. Furthermore, ImageNet features focus on texture and edges of macroscopic objects, which are semantically disjoint from the stochastic, texture-less patterns of neural oscillations. Our TS-FID utilizes a spectral feature extractor specifically designed to respect the physical meaning of frequency bands.

▷ **The phase-shift problem.** Evaluating FID directly on raw time-domain samples is equally problematic due to the phase-shift sensitivity. Two EEG signals can represent the exact same underlying oscillatory state but be slightly shifted in phase. A standard Euclidean-based feature extractor would treat these as distinct, penalizing the model for valid stochastic phase variations that are inherent in spontaneous brain activity. By mapping signals to the spectral domain before feature extraction, our TS-FID metric becomes phase-invariant while strictly penalizing deviations in power spectral density and rhythmic structure, aligning better with how clinicians interpret EEG based on rhythms and frequency content rather than raw voltage alignment.

# B. Implementation Details

In this section, we provide a comprehensive overview of the JET architecture and the exact hyperparameters used to reproduce our results.

## B.1. Model Architecture

As illustrated in Figure 7, the backbone of JET is a Transformer-based architecture designed to process continuous EEG signals. The pipeline proceeds as follows:

**Patchify & Tokenization:** The input 16-channel raw EEG segments ($\mathbf{x} \in \mathbb{R}^{C \times T}$) are first divided into fixed-size patches. These patches are linearly projected into a latent embedding space, effectively converting the continuous signal into a sequence of tokens.

**Transformer Encoder:** The tokenized sequence, combined with additive noise (as per the Flow Matching formulation), is processed by a stack of $L$ Transformer blocks. Each block consists of a Multi-Head Self-Attention (MHSA) mechanism followed by a Feed-Forward Network (FFN), with Layer Normalization (LayerNorm) and residual connections applied at each sub-layer. This design allows the model to capture global temporal dependencies and inter-channel correlations.

**Output Projection:** The processed tokens are unpatched and projected back to the original signal dimension to produce the predicted EEG waveform.

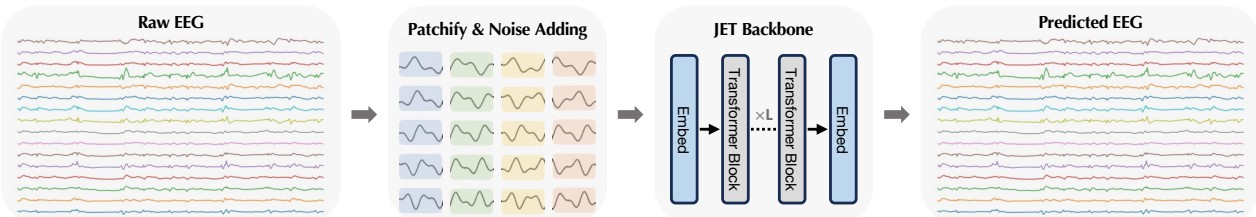

*Figure 7.* **Illustration of JET architecture.** The model processes raw EEG via a patch-based Transformer backbone. Input signals are tokenized, processed through stacked self-attention layers to capture spatiotemporal dependencies, and reconstructed to predict the target vector field.

## B.2. Training Configuration

Detailed hyperparameters for implementation in Table 9.

| Optimizer | Learning Rate | Label Drop | Batch Size | Epoch | Warmup | Betas | EMA | $P_{\text{train}}(\mu, \sigma)$ | Min Time ($t_\epsilon$) |
|---|---|---|---|---|---|---|---|---|---|
| AdamW | $5 \times 10^{-5}$ | 0.1 | 256 | 200 | 5 | $(0.9, 0.95)$ | 0.9999 | $(-0.8, 0.8)$ | $5 \times 10^{-2}$ |

*Table 9.* **Implementation details.** The values denote the hyperparameter configuration used in model training and inference.

## B.3. Computational Efficiency Analysis

To demonstrate the practical deployability of JET for large-scale EEG generation, we report the computational resource consumption, training throughput, and inference latency. All metrics reported below were measured on a single NVIDIA B200 GPU. The model configuration follows the standard setups with a patch size of $P = 200$.

**Training Efficiency and Tokenization.** Despite leveraging a Transformer backbone, JET achieves remarkable computational efficiency due to its aggressive patch-based tokenization. For the TUAB dataset ($T = 2000$), the input signal is projected into a concise sequence of only $L = T/P = 10$ tokens, and for other two datasets ($T = 1000$), $L = T/P = 5$. As shown in Table 10, training with a large batch size of 256 consumes approximately 21.5 GB of VRAM for TUAB and only 12.5 GB for TUEV/TUSZ. The training throughput is highly efficient, requiring only 1.16 seconds per step (256 samples) for the longest sequences.

**Inference Latency: Faster-than-Real-Time Generation.** Inference speed is critical for clinical applications. For TUAB, generating a batch of 32 samples takes 4.78 seconds. This translates to a per-sample latency of approximately 149 ms. Given

that the generated signal represents 10 seconds of biological time, JET achieves a generation speed roughly $67\times$ faster than real-time, enabling rapid offline data augmentation and near-instantaneous scenario simulation.

| Dataset | Input Shape $(C \times T)$ | Tokens $(T/P)$ | Params (M) | Peak VRAM (BS=256) | Train Speed (s/step) | Inf. Time (BS=32) | Per-Sample Latency |
|---|---|---|---|---|---|---|---|
| **TUAB** | $16 \times 2000$ | 10 | 129.9 | 21.5 GB | 1.16 s | 4.78 s | 149 ms |
| **TUEV / TUSZ** | $16 \times 1000$ | 5 | 129.9 | 12.5 GB | 0.46 s | 2.86 s | 89 ms |

*Table 10.* **Computational resource consumption.** We report the model parameters, peak memory usage, training speed (seconds per step @ Batch 256), and inference latency (total time for Batch 32 / per-sample latency). TUEV and TUSZ share identical input dimensions and thus exhibit similar performance.

# C. Theoretical Motivation of Principled Constraints

The constraint terms in Sec. 3.3 are motivated by well-established statistical and biophysical properties of scalp EEG recordings. We briefly summarize the rationale behind each component.

**Laplacian prior for reconstruction ($\mathcal{L}_{\textbf{recon}}$).** EEG measurements are frequently corrupted by impulsive artifacts arising from muscle activity or electrode motion, yielding heavy-tailed, non-Gaussian residuals (Roberts, 2000; Buzsáki & Mizuseki, 2014). An $L_2$ objective corresponds to a Gaussian error model and tends to oversmooth sharp transients. We therefore adopt an $L_1$ reconstruction loss, which is more robust to outliers and better preserves clinically relevant waveform discontinuities such as epileptiform spikes.

**Theorem.** Let $x \in \mathbb{R}^T$ be the ground truth signal and $\hat{x}$ be the estimate. If the residual noise $\epsilon = x - \hat{x}$ follows an i.i.d. Laplacian distribution with location parameter 0 and scale parameter $b$, i.e., $p(\epsilon) = \frac{1}{2b} \exp(-\frac{|\epsilon|}{b})$, then minimizing the $L_1$ norm is equivalent to the Maximum Likelihood Estimation of $x$.

**Proof.** The likelihood function for the sequence is given by

$$\mathcal{L}(\hat{x}|x) = \prod_{t=1}^{T} \frac{1}{2b} \exp\left(-\frac{|x_t - \hat{x}_t|}{b}\right). \tag{9}$$

Taking the negative log-likelihood:

$$-\log \mathcal{L}(\hat{x}|x) = \sum_{t=1}^{T} \frac{|x_t - \hat{x}_t|}{b} + T \log(2b) \propto \|x - \hat{x}\|_1 + C. \tag{10}$$

Minimizing the log-likelihood is thus strictly equivalent to minimizing the $L_1$ distance. Unlike the $L_2$ norm (Gaussian prior), which estimates the conditional mean, the $L_1$ norm estimates the conditional median, providing robustness against impulsive outliers.

**Statistical consistency ($\mathcal{L}_{\textbf{cons}}$).** EEG amplitude statistics vary substantially across time and brain states (Kaplan et al., 2005). In continuous flow-based generation, unconstrained dynamics may drift in mean or variance during integration, resulting in unstable trajectories. The consistency term regularizes the learned flow by matching first- and second-order statistics, anchoring generated samples within physiologically plausible amplitude ranges.

**Theorem.** Let $\mathcal{P}_{data}$ be the manifold characterized by the moment set $\mathcal{M} = \{(\mu, \sigma) \mid \mu \in \mathbb{R}, \sigma \in \mathbb{R}^+\}$. Minimizing $\mathcal{L}_{cons}$ projects the generated distribution $\mathcal{Q}$ onto $\mathcal{M}$ by minimizing the Maximum Mean Discrepancy with respect to the polynomial kernel $k(x, y) = (1 + \langle x, y \rangle)^2$.

**Proof.** Recall that the Maximum Mean Discrepancy (MMD) between distributions $\mathcal{P}$ and $\mathcal{Q}$ in a Reproducing Kernel Hilbert Space (RKHS), $\mathcal{H}$ is defined as the distance between their mean embeddings:

$$\text{MMD}^2(\mathcal{P}, \mathcal{Q}) = \|\mu_\mathcal{P} - \mu_\mathcal{Q}\|_\mathcal{H}^2 = \|\mathbb{E}_{x \sim \mathcal{P}}[\phi(x)] - \mathbb{E}_{\hat{x} \sim \mathcal{Q}}[\phi(\hat{x})]\|_\mathcal{H}^2, \tag{11}$$

where $\phi(\cdot)$ is the feature map associated with the kernel $k(x, y) = \langle \phi(x), \phi(y) \rangle_\mathcal{H}$.

Consider the second-order polynomial kernel $k(x, y) = (c + x^\top y)^2$ with bias $c = 1$. Expanding this kernel gives:

$$k(x, y) = 1 + 2x^\top y + (x^\top y)^2 = 1 + 2\sum_{i=1}^{T} x_i y_i + \sum_{i,j=1}^{T} (x_i x_j)(y_i y_j). \tag{12}$$

By inspection, the explicit feature map $\phi(x)$ decomposes into zeroth, first, and second-order terms:

$$\phi(x) = \left[1, \sqrt{2}x, \text{vec}(x \otimes x)\right]^\top. \tag{13}$$

Minimizing the MMD objective requires matching the expectations of these feature components:

$$\mathbb{E}_\mathcal{P}[\phi(x)] = \mathbb{E}_\mathcal{Q}[\phi(\hat{x})] \implies \begin{cases} \mathbb{E}[x] = \mathbb{E}[\hat{x}] & \text{(First Moment)} \\ \mathbb{E}[xx^\top] = \mathbb{E}[\hat{x}\hat{x}^\top] & \text{(Second Moment)} \end{cases} \tag{14}$$

Our proposed $\mathcal{L}_{\text{cons}}$ minimizes the $L_1$ divergence of the mean $\mu$ (first moment) and standard deviation $\sigma$ (derived from the diagonal of the second central moment). While the RKHS norm is induced by the $L_2$ metric in feature space, we employ the $L_1$ norm for robustness against outliers in the statistics estimates. Thus, $\mathcal{L}_{\text{cons}}$ serves as a tractable approximation to minimizing the MMD with a quadratic kernel, effectively projecting the generative flow onto the correct statistical manifold defined by $\mathcal{M}$. $\square$

**Spectral regularity ($\mathcal{L}_{\text{tv}}$).** Scalp EEG reflects cortical field potentials filtered through volume conduction, producing inherently smooth and band-limited observations. Spurious high-frequency fluctuations are thus more likely attributable to measurement noise than coherent neural activity. A temporal total variation penalty suppresses such artifacts while retaining structured oscillatory dynamics.

**Theorem.** Consider the local regularization of signal gradients $g_t = \nabla x_t$ given noisy observations $y_t$. The Total Variation penalty ($\|\nabla x\|_1$) acts as a non-linear soft-thresholding filter that suppresses small fluctuations (noise) while preserving large discontinuities (spikes), whereas Tikhonov regularization ($\|\nabla x\|_2^2$) applies a linear scaling that degrades all frequency components uniformly.

**Proof.** Let the observed gradient at time $t$ be $y_t$. We analyze the proximal operators associated with $L_1$ regularization on the gradient. The objective $\min_g \frac{1}{2}(y_t - g)^2 + \lambda|g|$ is solved by the soft-thresholding operator $\mathcal{S}_\lambda$:

$$\hat{g}_{TV} = \mathcal{S}_\lambda(y_t) = \text{sign}(y_t) \cdot \max(|y_t| - \lambda, 0). \tag{15}$$

The operator $\mathcal{S}_\lambda$ defines a dead zone in the interval $[-\lambda, \lambda]$. For noise-induced gradients where $|y_t| \leq \lambda$, the output is exactly zero ($\hat{g}_{TV} = 0$), enforcing piecewise smoothness. Crucially, for neural spikes where $|y_t| > \lambda$, the gradient is preserved (shifted by $\lambda$ but non-zero).

Thus, the $L_1$ norm on gradients is the unique convex regularizer that disentangles signal from noise based on magnitude sparsity, creating a non-linear filter essential for recovering the impulsive morphology of EEG without the spectral smearing characteristic of linear filters.

**Morphological alignment ($\mathcal{L}_{\text{corr}}$).** Absolute EEG amplitudes can vary considerably across subjects due to differences in impedance and sensor placement. Clinical interpretation, however, depends primarily on waveform morphology and phase structure rather than global scaling (Oppenheim & Lim, 2005). Pearson correlation provides a scale-invariant alignment measure, encouraging preservation of pathological waveform patterns independent of amplitude variation.

**Theorem.** Let $\mathcal{S}^{T-1} = \{u \in \mathbb{R}^T : \|u\|_2 = 1\}$ be the unit hypersphere. Define the projection operator $\Pi : \mathbb{R}^T \setminus \{0\} \to \mathcal{S}^{T-1}$ that maps a centered signal to the hypersphere as $\Pi(x) = \frac{x-\bar{x}}{\|x-\bar{x}\|_2}$. Maximizing the Pearson correlation $\rho(x, \hat{x})$ is equivalent to minimizing the squared Chordal distance between the projected points $\Pi(x)$ and $\Pi(\hat{x})$.

**Proof.** The Pearson correlation coefficient is defined as the inner product of the standardized vectors (z-scores). In geometric terms, this is the cosine of the angle $\theta$ between the centered, unit-norm vectors:

$$\rho(x, \hat{x}) = \langle \Pi(x), \Pi(\hat{x}) \rangle = \cos(\theta). \tag{16}$$

Consider the squared Euclidean distance (Chordal distance) between the projections on the hypersphere:

$$\mathcal{D}^2(\Pi(x), \Pi(\hat{x})) = \|\Pi(x) - \Pi(\hat{x})\|_2^2 \tag{17}$$
$$= \langle \Pi(x) - \Pi(\hat{x}), \Pi(x) - \Pi(\hat{x}) \rangle \tag{18}$$
$$= \|\Pi(x)\|_2^2 + \|\Pi(\hat{x})\|_2^2 - 2\langle \Pi(x), \Pi(\hat{x}) \rangle. \tag{19}$$

Since $\Pi(x)$ and $\Pi(\hat{x})$ lie on the unit hypersphere, $\|\Pi(x)\|_2^2 = \|\Pi(\hat{x})\|_2^2 = 1$. The equation simplifies to:

$$\mathcal{D}^2(\Pi(x), \Pi(\hat{x})) = 2 - 2\rho(x, \hat{x}) = 2(1 - \rho(x, \hat{x})). \tag{20}$$

Therefore, minimizing the loss $\mathcal{L}_{\text{corr}} = 1 - \rho(x, \hat{x})$ is equivalent to minimizing $\frac{1}{2}\mathcal{D}^2(\Pi(x), \Pi(\hat{x}))$. This proves that the objective optimizes the alignment of signal shapes on the statistical manifold $\mathcal{S}^{T-1}$, remaining strictly invariant to any affine transformation $x \mapsto \alpha x + \beta$ (for $\alpha > 0$) that would otherwise disrupt standard Euclidean metrics.

**Summary.** Together, these terms provide complementary regularization of the learned flow along robustness, statistical stability, and spatiotemporal structure.

# D. More ablation and analysis

In this section, we present a systematic ablation study examining the effect of loss weighting and noise initialization on the performance of JET. All experiments follow the standard training protocol described in Appendix B, and TS-FID scores are reported on a held-out evaluation set.

## D.1. Grid Search over Constraint Weights

To assess the robustness of JET with respect to its training objective, we perform a grid search over the weights of the proposed constraints ($\mathcal{L}_{\text{cons}}$, $\mathcal{L}_{\text{tv}}$, $\mathcal{L}_{\text{corr}}$). Each configuration is evaluated under two initialization regimes: Gaussian noise and deterministic zero noise. Table 11 summarizes the resulting TS-FID scores across all settings.

| Constraint Configuration | | | | Gaussian Noise (TS-FID ↓) | | | Zero Noise (TS-FID ↓) | | |
|---|---|---|---|---|---|---|---|---|---|
| $\mathcal{L}_{\text{recon}}$ | $\mathcal{L}_{\text{cons}}$ | $\mathcal{L}_{\text{tv}}$ | $\mathcal{L}_{\text{corr}}$ | TUAB | TUEV | TUSZ | TUAB | TUEV | TUSZ |
| 1 | 0 | 0 | 0 | 188.39 | 244.50 | 237.75 | 1842.03 | 1939.22 | 1954.97 |
| 1 | 0 | 0 | 0.05 | 195.92 | 237.29 | 313.99 | 1718.31 | 1961.80 | 1978.40 |
| 1 | 0 | 0 | 0.1 | 197.51 | 247.45 | 188.99 | 1767.32 | 1886.08 | 1953.35 |
| 1 | 0 | 0.05 | 0 | 191.31 | 241.58 | 239.53 | 1743.47 | 1948.80 | 1939.33 |
| 1 | 0 | 0.05 | 0.05 | 233.76 | 246.19 | 229.07 | 1694.69 | 1959.27 | 1989.13 |
| 1 | 0 | 0.05 | 0.1 | 199.82 | 286.60 | 226.53 | 1715.86 | 1957.67 | 1962.09 |
| 1 | 0 | 0.1 | 0 | 192.15 | 270.70 | 240.22 | 1723.67 | 1991.95 | 1935.46 |
| 1 | 0 | 0.1 | 0.05 | 207.34 | 247.89 | 343.59 | 1772.73 | 1970.55 | 1974.62 |
| 1 | 0 | 0.1 | 0.1 | 225.87 | 243.06 | 208.81 | 1780.34 | 1977.18 | 1937.11 |
| 1 | 1 | 0 | 0 | 227.40 | 503.42 | 268.50 | 1626.17 | 1949.73 | 1967.78 |
| 1 | 1 | 0 | 0.05 | 169.68 | 246.33 | 176.80 | 1736.93 | 1978.98 | 1983.89 |
| 1 | 1 | 0 | 0.1 | 171.94 | 247.34 | 187.57 | 1702.53 | 1970.76 | 1928.95 |
| 1 | 1 | 0.05 | 0 | 277.99 | 466.42 | 310.52 | 1575.67 | 1957.54 | 2042.99 |
| 1 | 1 | 0.05 | 0.05 | 229.02 | 268.16 | 176.53 | 1670.42 | 1924.30 | 1992.98 |
| 1 | 1 | 0.05 | 0.1 | **188.27** | 238.34 | 194.64 | 1615.32 | 1914.12 | 1979.59 |
| 1 | 1 | 0.1 | 0 | 255.19 | 505.52 | 249.40 | 1576.96 | 1996.72 | 1878.64 |
| 1 | 1 | 0.1 | 0.05 | 242.19 | 286.30 | **151.27** | 1600.90 | 1924.06 | 1958.24 |
| 1 | 1 | 0.1 | 0.1 | 221.93 | **235.86** | 190.21 | 1640.63 | 1964.93 | 1966.93 |
| 1 | 3 | 0 | 0 | 317.35 | 678.60 | 439.85 | 1547.87 | 1988.45 | 1928.36 |
| 1 | 3 | 0 | 0.05 | 269.22 | 306.93 | 264.44 | 1678.44 | 1989.93 | 1996.33 |
| 1 | 3 | 0 | 0.1 | 207.99 | 285.40 | 223.81 | 1678.95 | 1971.70 | 1972.94 |
| 1 | 3 | 0.05 | 0 | 329.94 | 762.14 | 283.97 | 1491.51 | 1980.94 | 1930.96 |
| 1 | 3 | 0.05 | 0.05 | 287.82 | 338.94 | 247.20 | 1527.20 | 1981.80 | 1966.88 |
| 1 | 3 | 0.05 | 0.1 | 214.74 | 239.22 | 229.24 | 1686.33 | 1987.61 | 1932.73 |
| 1 | 3 | 0.1 | 0 | 313.82 | 615.91 | 336.89 | 1458.72 | 1993.36 | 1952.17 |
| 1 | 3 | 0.1 | 0.05 | 284.01 | 271.63 | 236.81 | 1491.14 | 1981.70 | 1937.84 |
| 1 | 3 | 0.1 | 0.1 | 214.89 | 258.06 | 231.95 | 1598.48 | 1984.90 | 1888.31 |

*Table 11.* **Ablation results on gaussian noise and zero noise settings.** Values denote constraint weights.

▷ **The Limitation of Zero-Noise Initialization.** The right half of Table 11 provides a strong negative control that isolates the effect of the base distribution. Across all constraint configurations, including settings with large weights on smoothness or correlation, deterministic zero-noise initialization consistently leads to catastrophic TS-FID scores ($> 1500$). This indicates that the failure cannot be remedied through loss reweighting alone.

From a generative modeling perspective, initializing from a single deterministic point forces the model to expand a degenerate distribution into a highly multimodal EEG data distribution. Such a mapping is fundamentally ill-posed under continuous flow-based formulations, as the learned vector field must simultaneously create diversity and preserve structure. In contrast, a stochastic base distribution with sufficient dispersion provides the necessary degrees of freedom for the flow to cover the data support. These results empirically demonstrate that a non-degenerate, high-entropy base distribution is essential for learning a valid transport map in flow matching, particularly for complex physiological signals.

▷ **Sensitivity to Constraint Design.** We next analyze the Gaussian noise regime (left half of Table 11), which reveals clear and interpretable patterns in how different constraints influence generative performance. These trends highlight the interaction between optimization objectives and the statistical structure of EEG signals:

(1) **Energy Regularization Exhibits a Clear Optimum.** Performance as a function of the energy consistency weight $\mathcal{L}_{cons}$ follows a concave trend. Removing this constraint ($\mathcal{L}_{cons} = 0$) leads to inferior performance, indicating insufficient regulation of signal amplitude statistics. Moderate weighting ($\mathcal{L}_{cons} = 1$) achieves the best balance. However, overly strong enforcement ($\mathcal{L}_{cons} = 3$) significantly degrades performance, suggesting that excessive moment matching constrains the vector field too rigidly and suppresses legitimate data-driven variability of neural dynamics.

(2) **Understanding the Role of Structural Constraints in EEG Flow Modeling.** The effectiveness of the proposed constraints varies systematically across datasets. For TUSZ, which contains high-amplitude, highly non-stationary seizure activity, incorporating principled constraints yields a substantial performance improvement of approximately 36%. In contrast, gains on TUAB, which is dominated by subtler background abnormalities, are more modest. This pattern indicates that explicit structural regularization is particularly important for modeling extreme, transient pathological events, where unconstrained objectives tend to collapse toward average behavior.

(3) **Complementarity of Smoothness and Correlation Constraints.** The strongest performance is consistently achieved when temporal smoothness ($\mathcal{L}_{tv}$) and correlation alignment ($\mathcal{L}_{corr}$) are applied jointly. Applying either constraint in isolation often fails to improve, and in some cases degrades, performance relative to the baseline. This suggests that structural constraints require stabilization of signal energy to be effective; without $\mathcal{L}_{cons}$, enforcing smoothness or waveform alignment can inadvertently distort the underlying power-law spectral structure.

(4) **Robustness of the Optimal Configuration.** A single configuration, $\mathcal{L}_{cons} = 1$, $\mathcal{L}_{tv} = 0.05/0.1$, and $\mathcal{L}_{corr} = 0.05/0.1$, consistently ranks among the top-performing settings across TUAB, TUEV, and TUSZ. Despite the substantial differences in spectral composition and temporal dynamics among these datasets, this configuration demonstrates stable performance, supporting its use as a robust default for EEG generative modeling.

## D.2. More Qualitative Visual Analysis

To provide further insight into the distributional behavior of the generated signals, we visualize the same analyses used in the ablation study. These visualizations offer qualitative evidence of how different noise priors affect distributional coverage and structure, illustrating how the choice of noise prior influences the ability of the learned flow to model the multi-modal data distribution.

▷ **Visualizing the Gaussian Advantage (Figures 8 – 11).** Under Gaussian noise initialization, the visualizations provide qualitative evidence that JET learns a generative flow that effectively spans the support of the EEG data distribution. Across amplitude, spectral, and temporal dimensions, the generated samples closely match the structure of real EEG signals, reflecting the model's ability to traverse the neural manifold in a continuous and stable manner. We analyze these behaviors along the three key challenges identified in Sec. 1:

(1) **Heavy-Tailed Amplitude Distributions (Fig. 8).** The amplitude histograms show strong alignment between the log-density of generated and real signals across the full dynamic range. JET preserves both the sharp central peak and the heavy-tailed decay of the distribution, indicating that the learned flow does not collapse toward a Gaussian regime. This behavior reflects accurate modeling of higher-order amplitude statistics, rather than convergence toward average energy levels. Consequently, rare but clinically meaningful high-amplitude events, including ictal discharges and transient bursts, remain well represented in the generated samples.

(2) **Power-Law Spectral Structure (Fig. 9, 10).** Spectral analyses show that generated signals faithfully preserve the power-law scaling behavior ($1/f^{\chi}$) across frequencies. Unlike baseline models that exhibit spectral bias and attenuate high-frequency components, JET maintains close alignment with the ground truth even in the $\beta$ and $\gamma$ bands ($> 20$ Hz). Furthermore, distinct oscillatory signatures, such as $\alpha$-band peaks around 10 Hz, are consistently reproduced. This indicates that the learned vector field preserves structured oscillatory components and band-specific dynamics, rather than merely matching a global spectral slope.

(3) **Non-Stationary Temporal Dynamics (Fig. 11).** Temporal visualizations demonstrate variance envelopes that evolve smoothly and non-uniformly over time. These fluctuations reflect genuine non-stationary state transitions, rather than artifacts of the generation process. At the same time, the absence of baseline drift or variance explosion indicates that the model maintains stable amplitude statistics over long horizons. Together, these observations suggest that JET successfully captures non-stationary temporal dynamics while remaining statistically well-behaved, a balance that is difficult to achieve with discretized or unconstrained generative formulations.

▷ **Visualizing Degenerate Behavior under Zero Noise (Figures 12 – 15).** In contrast to Gaussian noise initialization, the zero-noise setting exposes systematic failure modes induced by a degenerate base distribution. Eliminating stochasticity at initialization restricts the set of admissible transport trajectories, impairing the learned flow across distributional, spectral, and temporal dimensions. These behaviors are consistent across datasets and manifest as follows:

**(1) Distribution Collapse and Limited Amplitude Coverage (Fig. 12).** The amplitude histograms exhibit sharp, unnatural spikes with probability mass concentrated near zero, together with a pronounced reduction of variability in the tails. This behavior indicates that, when initialized from a deterministic point, the learned vector field converges toward a narrow region of the signal space. Consequently, the generative process fails to explore the full range of EEG amplitudes, producing samples with limited diversity and an absence of extreme events observed in real recordings.

**(2) Distorted Spectral Structure (Fig. 13).** Spectral analysis reveals substantial deviations from the ground-truth power-law behavior. In particular, the $1/f^\chi$ decay is flattened or inconsistently reproduced across frequencies. This suggests that, without a stochastic base distribution, the learned flow struggles to recover scale-free spectral organization, resulting in spectra that lack coherent structure across frequency bands.

**(3) Degenerate Temporal Evolution (Fig. 15).** Temporal visualizations further highlight the limitations of zero-noise initialization. Generated sequences exhibit either erratic variance fluctuations or near-constant trajectories over time. Both patterns indicate a failure to sustain coherent temporal evolution, resulting in signals that are either unstable or lack the structured, time-varying dynamics observed in real EEG recordings.

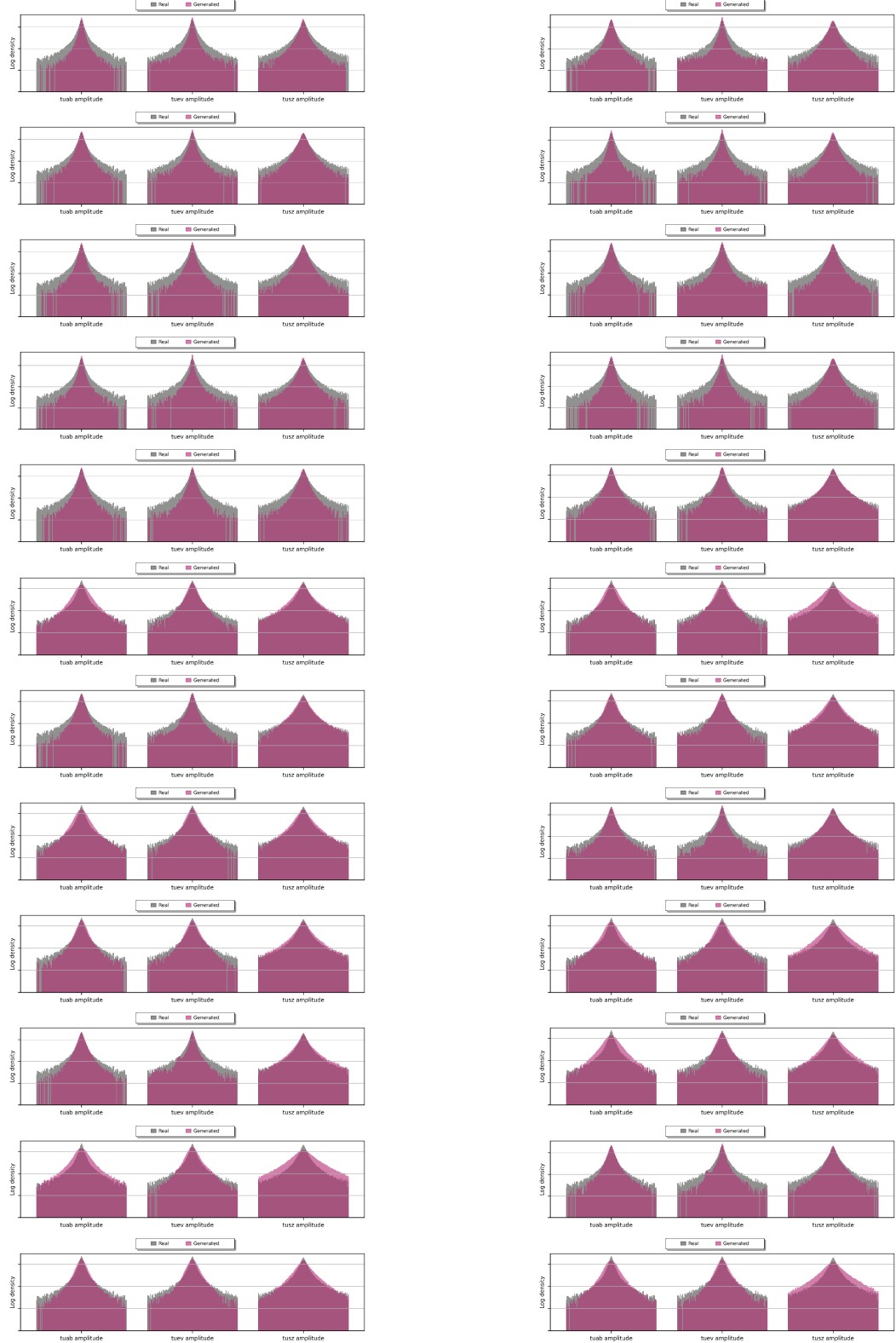

*Figure 8.* **Gaussian noise: amplitude distribution.** The generated signals closely match the ground truth log-density histograms, effectively capturing the heavy-tailed distribution of EEG amplitudes across all datasets.

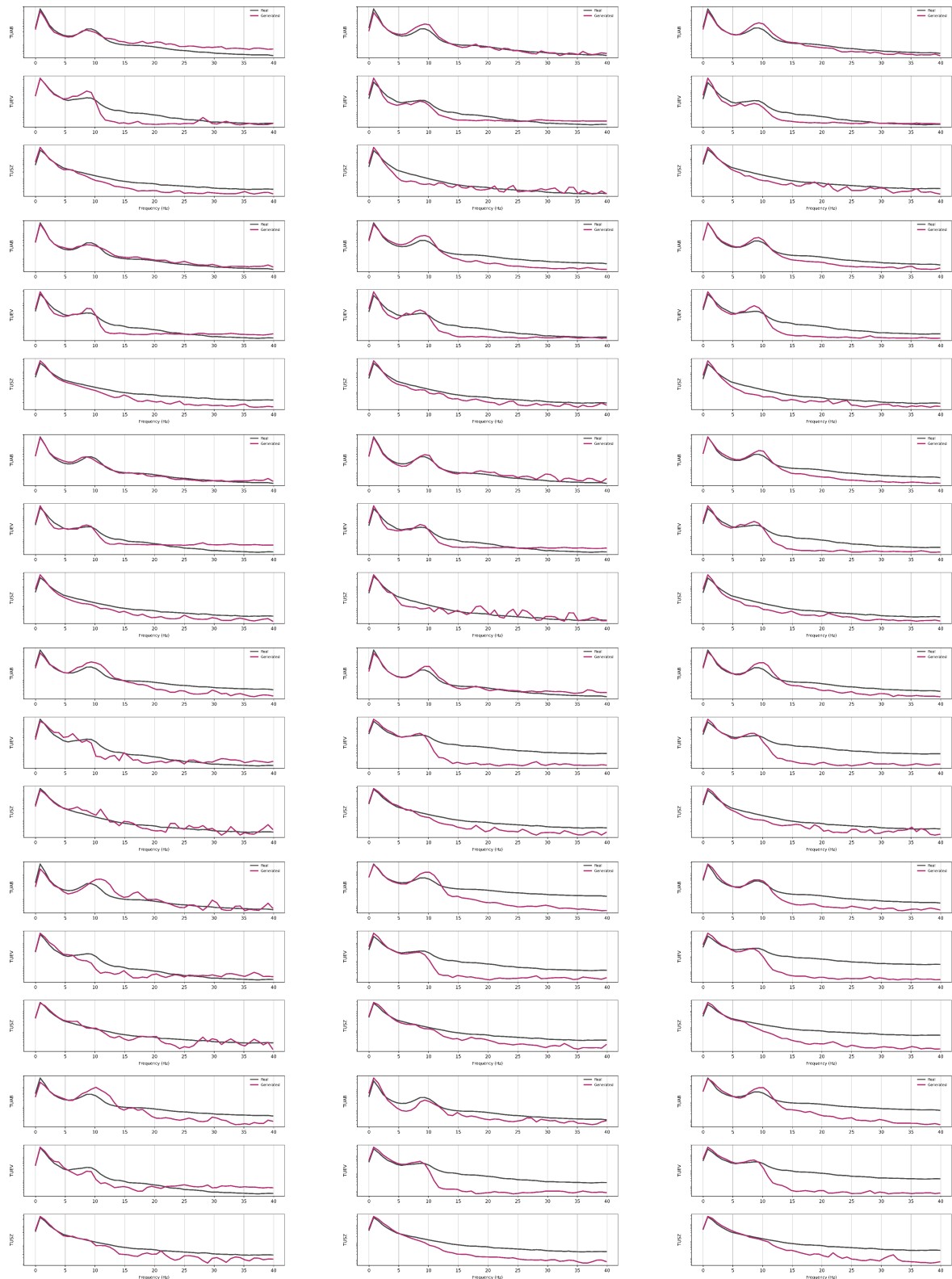

*Figure 9.* **Gaussian noise: power spectral density.** The model accurately reconstructs the $1/f$ spectral slope and distinct oscillatory peaks, demonstrating high spectral fidelity.

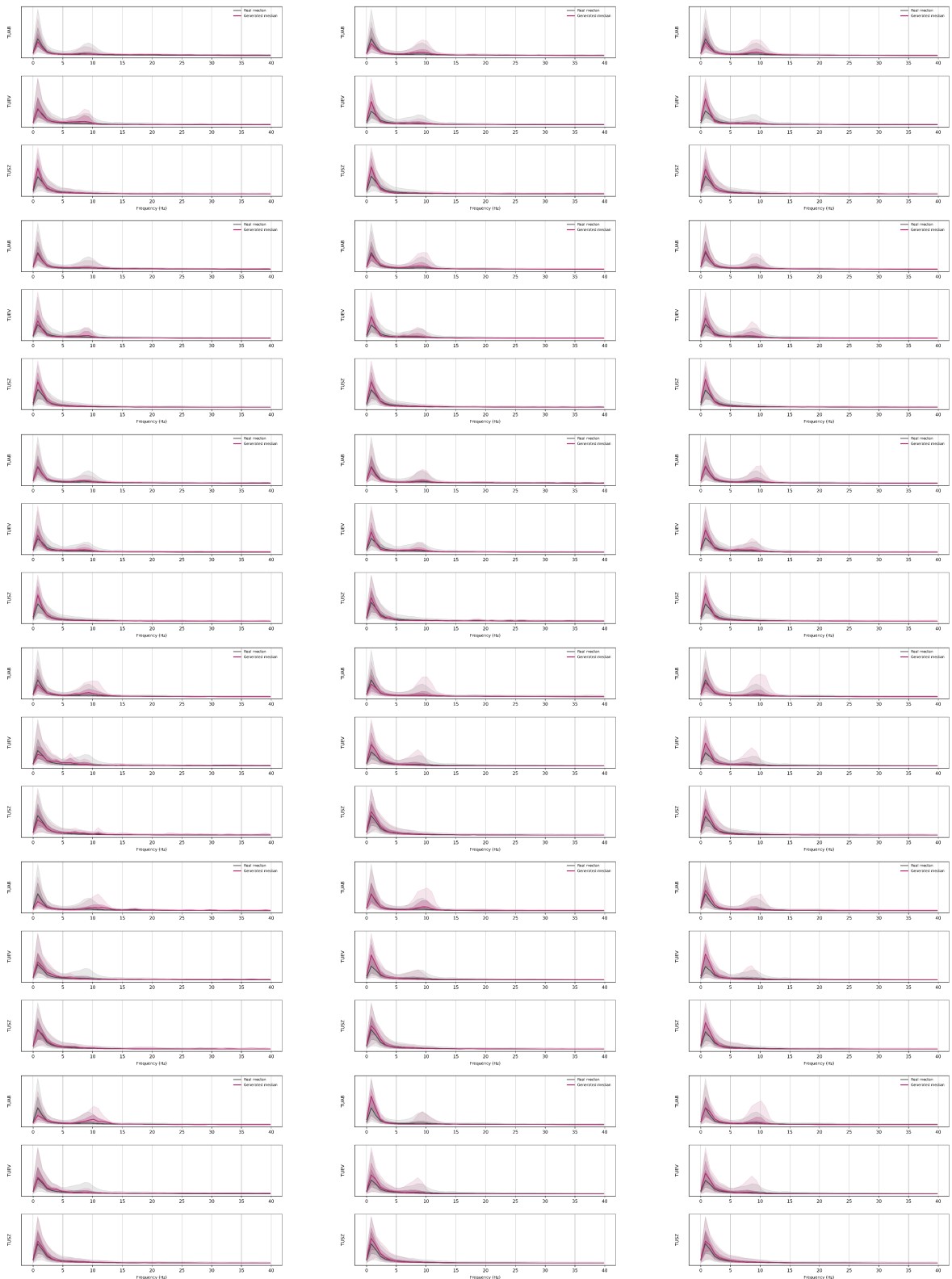

*Figure 10.* **Gaussian noise: spectral distribution stability.** The shaded percentile bands of the generated spectra overlap tightly with the real data, indicating that JET captures the full diversity of population-level variance.

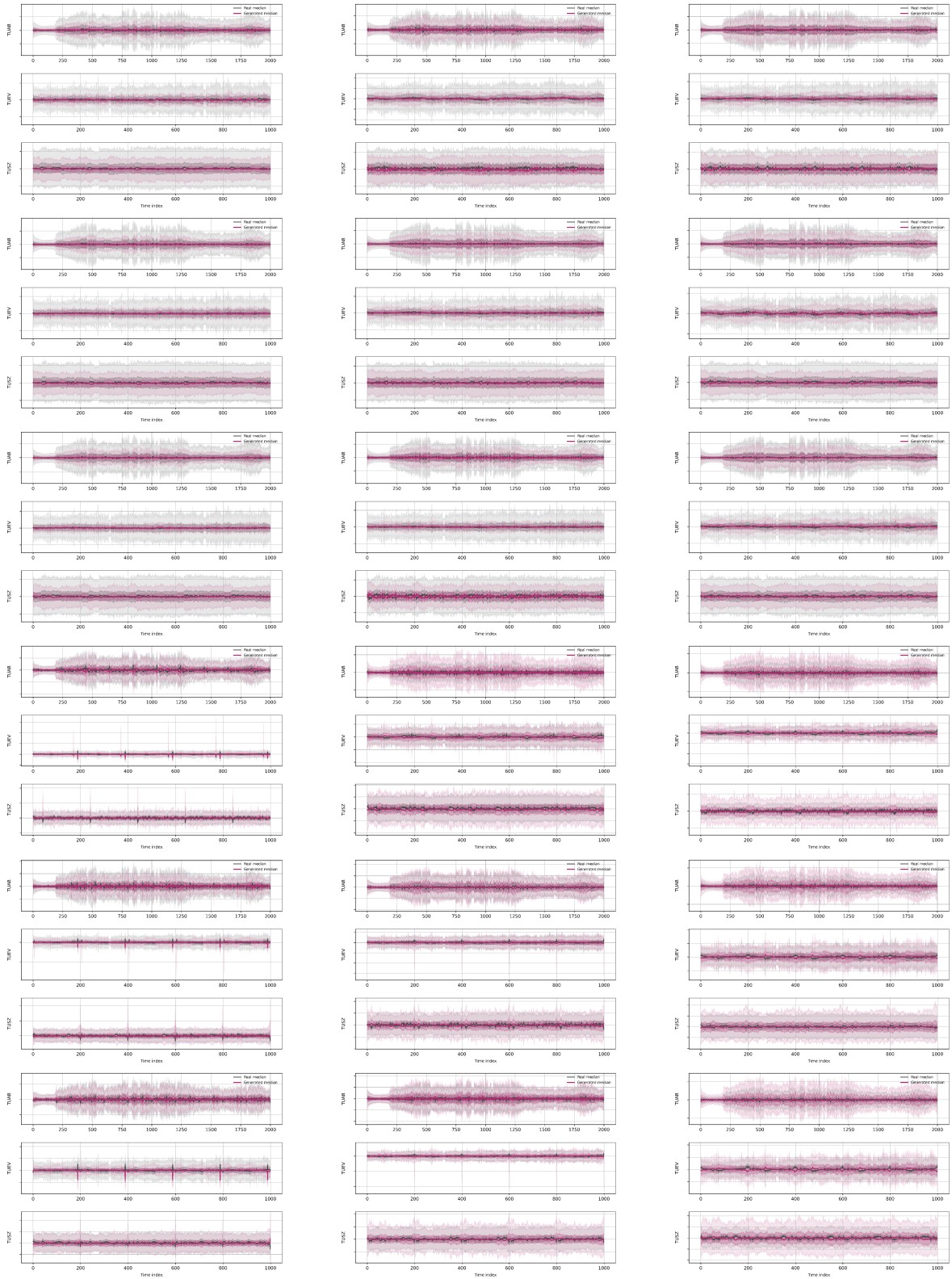

*Figure 11.* **Gaussian noise: temporal stationarity.** Z-normalized temporal envelopes exhibit stable, non-stationary variance dynamics that mirror the natural evolution of biological signals without drift.

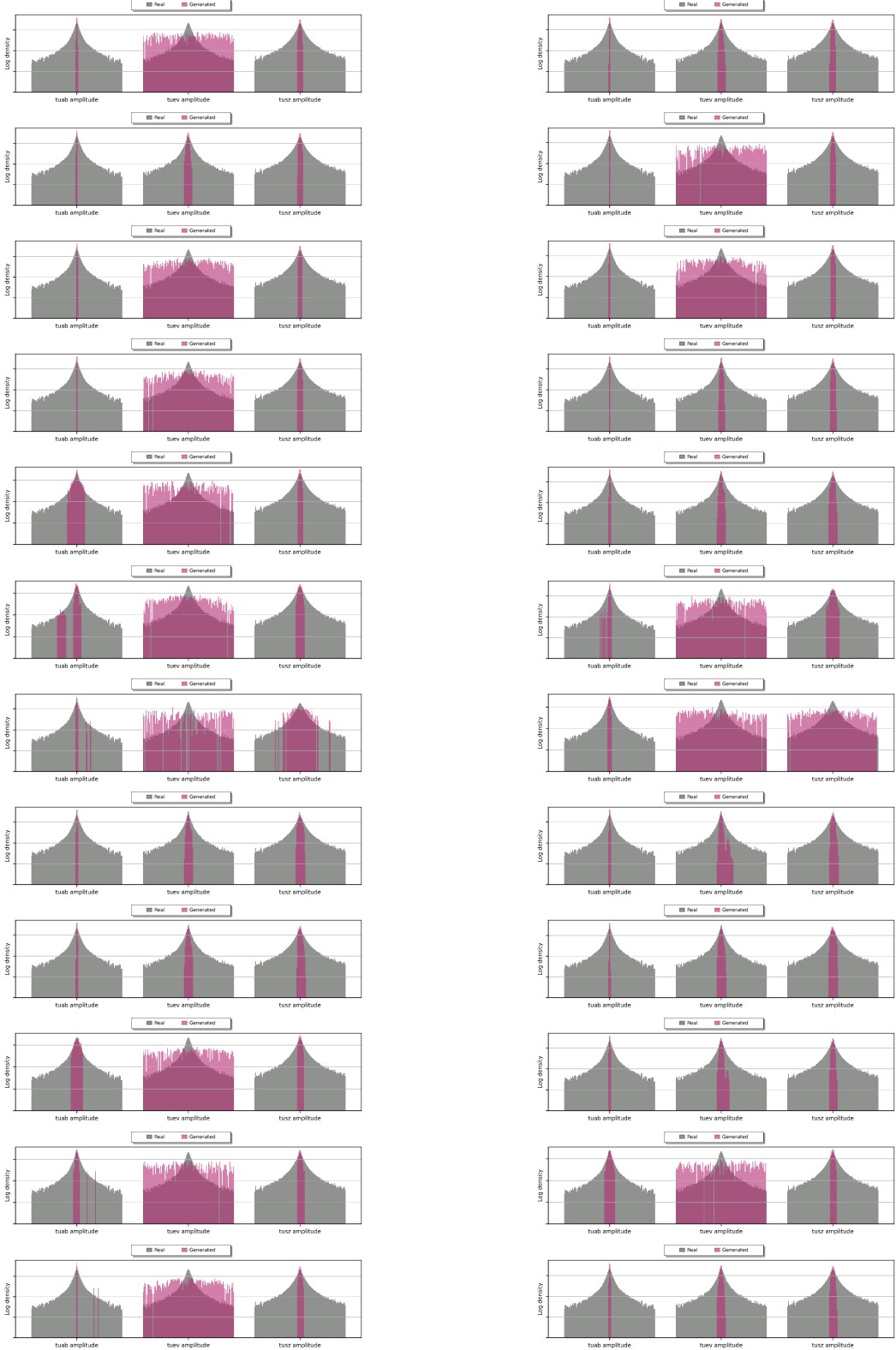

*Figure 12.* **Zero noise: amplitude distribution.** Initializing with zero noise leads to severe mode collapse, visualized as sharp, unnatural spikes in the histograms and a failure to cover the marginal distribution.

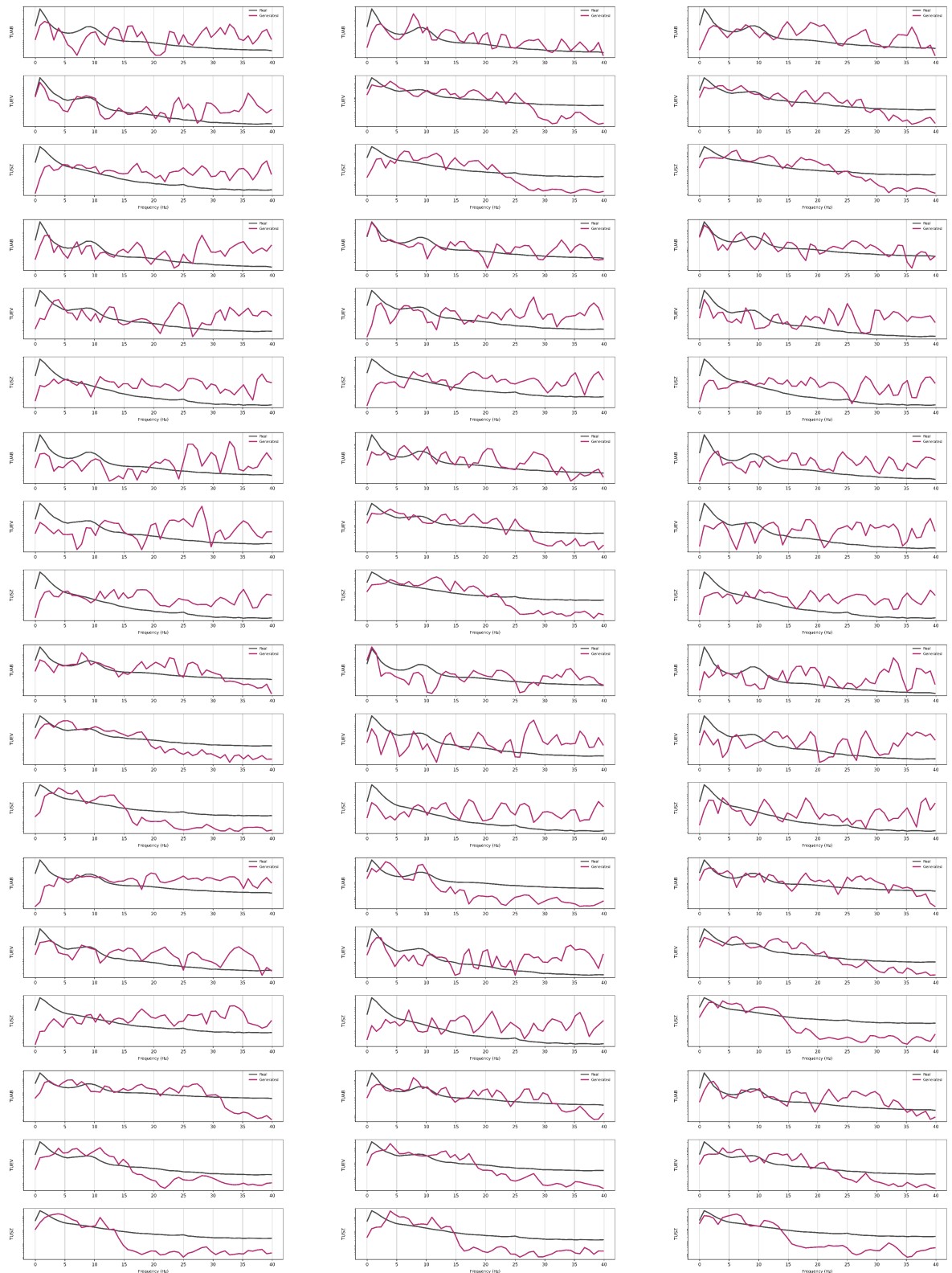

*Figure 13.* **Zero noise: power spectral density.** The spectra exhibit significant distortion, with flattened slopes and mismatched energy levels, confirming the inability to model scale-free dynamics from a singularity.

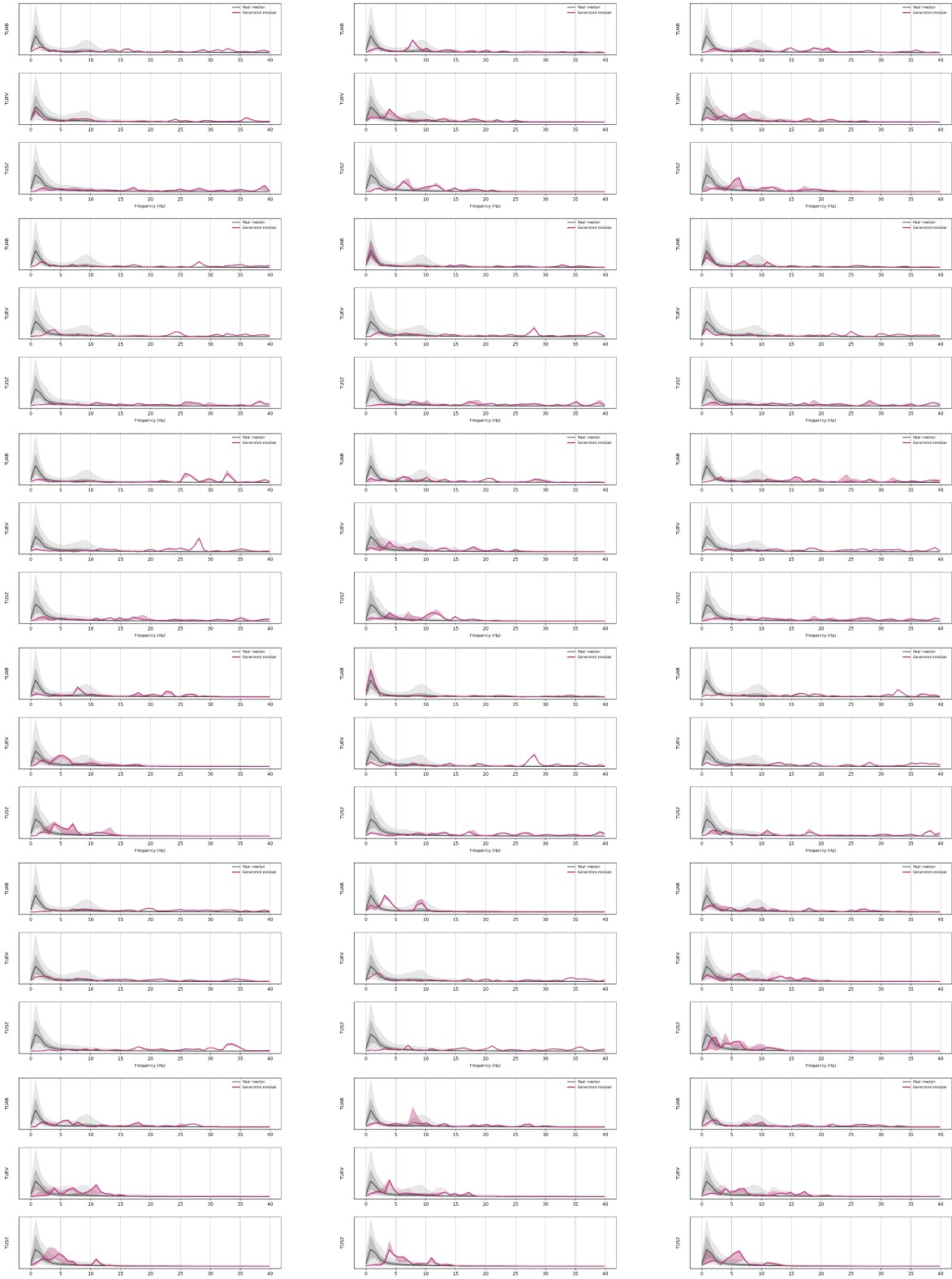

*Figure 14.* **Zero noise: spectral distribution stability.** The generated variance bands are either significantly collapsed (too narrow) or distorted, failing to represent the natural spectral diversity of the dataset.

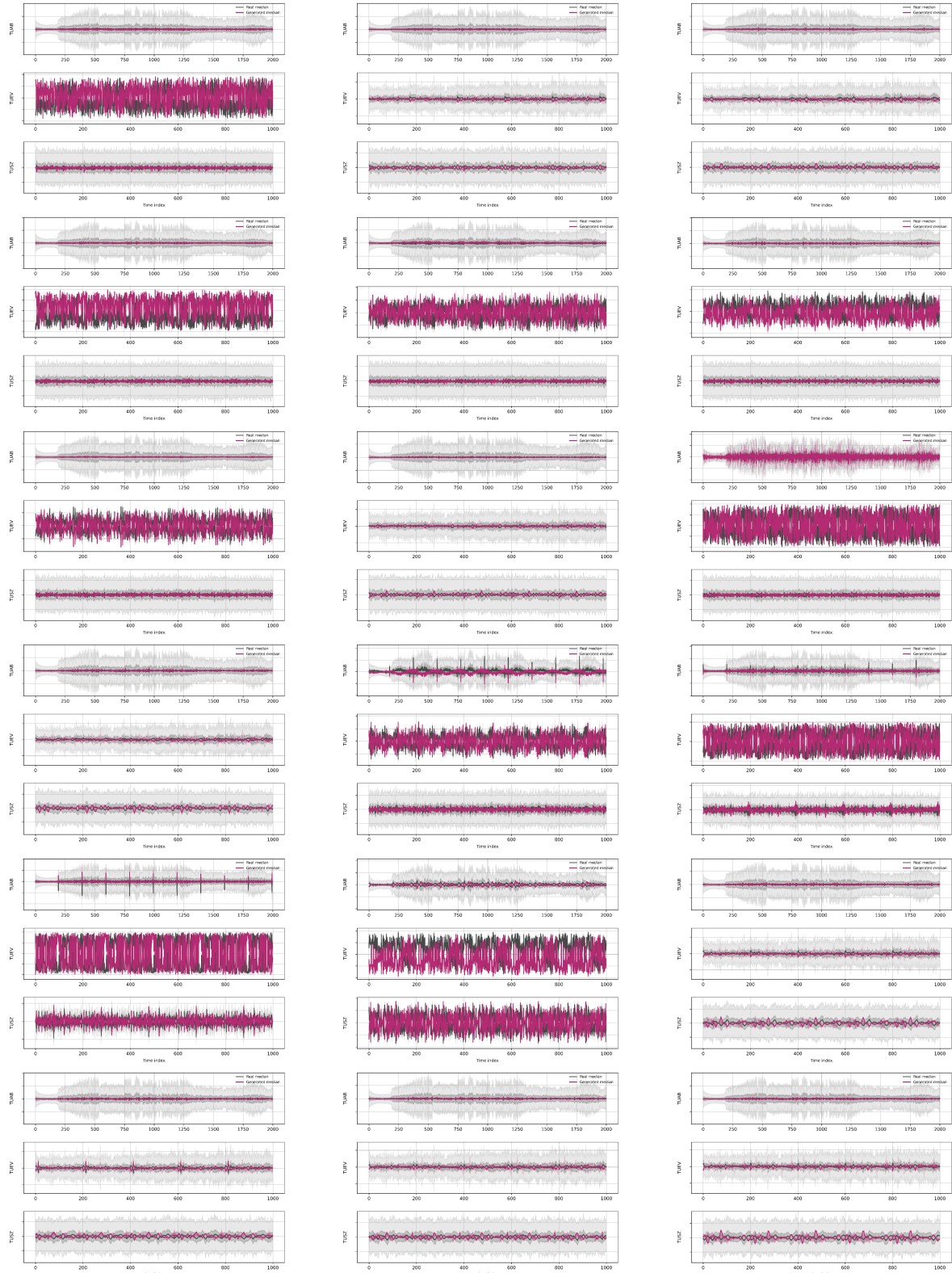

*Figure 15.* **Zero Noise: temporal stationarity.** The temporal traces display erratic fluctuations or flat-lining, reflecting the failure of the flow trajectory to evolve into valid temporal patterns.

