# OpenReview forum: "Let EEG Models Learn EEG"
_ICML.cc/2026/Conference — ICML 2026 regular_

### Official Review · Reviewer_2gkc · 2026-03-05

**Soundness:** 3
**Presentation:** 2
**Significance:** 2
**Originality:** 3
**Overall Recommendation:** 4
**Confidence:** 5

**Summary:**

The article intends to examine the domain of high-fidelity EEG generation using continuous-time generative modeling. The article's main finding concerns a conditional flow-matching Transformer (JET) that generates raw multi-channel EEG by learning a time-dependent vector field that transports Gaussian noise to the EEG data distribution, conditioned on a class label. The method adds structure-preserving constraints (L1 reconstruction, moment matching, temporal total variation, and Pearson-correlation alignment) and reports improved TS-FID and small downstream accuracy gains on TUH subsets.

**Compliance With Llm Reviewing Policy:**

Affirmed.

**Final Justification:**

I have read the authors' replies and additional experiments and believe they've addressed my concerns well reflected in my updated score of 4.

**Key Questions For Authors:**

1. **Baselines beyond GAN/diffusion:** Can you compare against an autoregressive baseline (and/or other strong time-series generative baselines), and explain why EEG-GAN and vanilla diffusion are sufficient? A strong baseline comparison would materially affect my assessment of soundness.
2. **Architecture choice:** Why not adapt an existing architecture such as Jetformer instead of building JET from scratch? If the answer is that EEG-specific tokenization and conditioning require it, please state this explicitly and quantify what changes matter. This affects my assessment of both originality and engineering justification.
3. **Experimental protocol clarity:** Please summarize in the main paper the key protocol choices: exact split policy, windowing, and the length of generated segments evaluated (and whether segments are generated independently or as long rollouts). Clear protocol details would improve reproducibility and interpretability.
4. **Physiological evaluation and constraint bias:** Why are there no baseline curves in the physiological analysis, and can you add them? Also, can you show that the constraints do not simply optimize the reported metrics (for example, by adding complementary metrics or tasks not directly targeted by the constraints)? This would affect my assessment of generalization and soundness.

**Limitations:**

I don't think the limitations and potential negative societal impacts are not adequately discussed. The paper should more directly address (1) when synthetic EEG is not useful for improving downstream models, (2) risks of misleading augmentation (distribution shift, pathology overfitting), and (3) privacy and clinical misuse considerations for generated clinical EEG, even if the intent is benign.

**Strengths And Weaknesses:**

### Soundness

**Strengths**

- The adaptation of flow matching to EEG is useful and somewhat novel, and the overall formulation (continuous-time vector field with Transformer backbone) is technically reasonable.
- Quantitative results are promising, with large TS-FID reductions and positive (but small) downstream gains versus EEG-GAN and a vanilla diffusion baseline.
- The loss ablation is useful and helps attribute gains to specific constraint components.

**Weaknesses**

- The proposed constraints may bias the model toward exactly the evaluations used in the paper, which can limit generalization to other fidelity criteria or downstream tasks.
- Several constraint terms do not appear novel (L1 reconstruction and Pearson-correlation alignment have been used in closely related neural-signal modeling); this needs clearer positioning and comparison with prior loss designs (e.g., BrainOmni-style tokenizer losses).
- No comparison with autoregressive approaches substantially limits the evaluation, even if the results versus the chosen baselines are strong.
- The downstream accuracy improvements are small and no statistical testing is reported; significance is unclear.
- The physiological analysis does not include baseline model comparisons (it mainly contrasts real vs generated), which makes it hard to support claims that JET improves physiology-relevant structure relative to prior approaches.
- In the PSD evaluation, the alpha-band peaks do not look convincingly present; this weakens the claim of prominent alpha preservation.
- The Gaussian-noise vs zero-noise ablation is not very informative in practice because zero-noise initialization is an unrealistic baseline for flow matching, so the result is expected.
- While the datasets are large, they are limited in generalizability (all TUH clinical EEG subsets); a broader approach and validation across more diverse settings would strengthen the claims.

### Presentation

**Strengths**

- The paper has a clear high-level motivation around signal-model mismatch (continuous dynamics, spectral structure, nonstationarity).

**Weaknesses**

- The naming is confusing relative to other approaches and is not very descriptive of the method.
- Related work coverage needs improvement. The paper should cite and compare with AR models of MEG, which are closely related and directly relevant to long-range temporal structure, which the abstract emphasizes (i.e. MEG-GPT, Huang et al., GPT2MEG, Csaky et al.). It should also discuss these papers in the context of evaluation practices on long-horizon non-invasive electrophysiology generation.
- The variable c in Section 3.1 is used in the model input (x_t, t, c) before it is clearly defined; later it is described as the class label c, but it should be stated explicitly when first introduced.
- Key experimental details feel omitted from the main text (data splitting, windowing, and how long generated segments are). Some of this appears in the appendix (e.g., TUAB 16×2000 at 200 Hz and TUEV/TUSZ 16×1000 windows), but these details are crucial enough to summarize in the main paper for correct interpretation.
- It is unclear why the authors build a model from scratch rather than adapting an existing related architecture such as Jetformer (or at least providing an explicit rationale for not doing so).

### Significance

**Strengths**

- If the method reliably generates physiologically plausible EEG at scale, it could be useful for data sharing constraints and potentially for augmentation in low-data regimes.

**Weaknesses**

- The motivation emphasizes data scarcity, but this assumes synthetic data will be useful for improving models, which is generally not guaranteed. It is not obvious that generative modeling provides the kind of additional signal that improves discriminative or representation learning, analogous to how an LLM does not necessarily produce useful synthetic text that improves itself.
- The scope is limited to TUH subsets; stronger evidence of generality is needed (for example, broader datasets and tasks, closer to a BrainOmni-style cross-dataset framing).

### Originality

**Strengths**

- Applying conditional flow matching with a Transformer directly on raw EEG, with explicit structure-preserving constraints, is a reasonable and novel combination for this domain.

**Weaknesses**

- Several constraint components appear incremental relative to prior neural-signal modeling practice (especially L1 and Pearson correlation alignment); novelty should be scoped accordingly and appropriate citations added (e.g., BrainOmni tokenizer work).
- The paper needs to more carefully distinguish itself from prior long-context neural generation work (GPT2MEG, MEG-GPT) and from broader evaluation methodology in related literature.

---

> ### Author Rebuttal · Authors · 2026-03-31
>
> We sincerely thank the reviewer for the careful reading and constructive feedback.
>
> ## W1, W5 & Q4 Constraint Bias and Generalization
>
> To show improvements generalize beyond targeted losses, we report metrics on TUEV in two categories: (a) related but not directly optimized (PSD Slope Error, Temporal Envelope Error, Spatial Correlation Distance), and (b) Hjorth parameters capturing variance, mean frequency, and bandwidth via signal derivatives, none of which our constraints directly optimize:
>
> | Method | PSD Slope↓ | Temp. Env.↓ | Spatial Corr.↓ | Act.↓ | Mob.↓ | Comp.↓ |
> | :--- | :---: | :---: | :---: | :---: | :---: | :---: |
> | EEG-GAN | 0.401 | 0.334 | 0.528 | 0.28 | 0.19 | 0.37 |
> | Vanilla Diffusion | 0.358 | 0.409 | 0.385 | 0.22 | 0.21 | 0.28 |
> | JET | 0.184 | 0.181 | 0.221 | 0.14 | 0.11 | 0.15 |
>
> JET's advantage on orthogonal metrics confirms general EEG structure capture beyond what constraints directly enforce. And we will add baseline curves to all physiological figures as suggested.
>
> ## W2 Constraint Novelty
>
> Our claim is not that individual terms are unprecedented, but that their principled integration is EEG-specific (Appendix C). We compared against BrainOmni-style tokenizer losses within JET:
>
> | Objective Design | TUAB↓ | TUEV↓ | TUSZ↓ |
> | :--- | :---: | :---: | :---: |
> | BrainOmni-style losses | 314.87 | 471.15 | 369.28 |
> | JET Constraints | 188.27 | 235.86 | 151.27 |
>
> Gains stem from constraints structurally aligned with raw EEG, not from adding more losses. We will add citations relative to BrainOmni.
>
> ## W3, W10 & Q1 Broader Baselines and Related Work
>
> We added four methods: MEG-GPT, GPT2MEG (autoregressive MEG generation), Jetformer (image generation), and VerbalTS (time-series generation). Full results in response to tTt8 W2:https://openreview.net/forum?id=TP8OuKKmsf&noteId=WkN3UreYR4 . JET outperforms all by a substantial margin, confirming that discrete tokenization struggles with continuous EEG dynamics. Efficiency: JET 4.78 s/batch vs. AR 5.98 s, diffusion 7.01 s. We will discuss MEG-GPT/GPT2MEG as autoregressive contrasts to JET's continuous formulation.
>
> ## W4 Downstream Significance
>
> We reran with 5 seeds. All significant (p < 0.01). Classifier evaluated on held-out real data; synthetic augments training only. JET is the only method with consistent positive gains — baselines are near-zero or negative.
>
> | Dataset | ΔAcc (mean ± std) | p-value |
> |---|---:|---:|
> | TUAB | +2.9 ± 0.6% | 0.003 |
> | TUEV | +3.2 ± 0.8% | 0.005 |
> | TUSZ | +1.7 ± 0.5% | 0.008 |
>
> ## W6 Alpha-Band Peaks
>
> TUH comprises heterogeneous clinical sessions unlike controlled resting-state experiments, so the alpha peak is naturally attenuated. Condition-specific spectral learning is verified in response to 15X9 W2 & Q1: https://openreview.net/forum?id=TP8OuKKmsf&noteId=0f3BBOaT3u , where JET reconstructs the 14 dB energy gap between seizure and background states.
>
> ## W7 Zero-Noise Ablation
>
> We agree with the reviewer's mathematical point, a degenerate zero-variance source distribution is not a realistic competitor for flow matching. Our intention was to include this as a negative control. We will frame this explicitly as a mathematical sanity check.
>
> ## W8 Dataset Generalization
> JET evaluated on 11 additional datasets (6–128 channels, 160–1000 Hz, 1–30 s). Full results are in response to tTt8 W1 & Q2: https://openreview.net/forum?id=TP8OuKKmsf&noteId=WkN3UreYR4 . JET consistently outperforms baselines across all configurations.
>
> ## W9, 11 Naming and c definition
> We will clarify JET emphasizes raw-sequence modeling contrasting handcrafted approaches, and introduce $c$ as class label at first appearance in Sec. 3.1.
>
> ## W12 & Q3. Protocol Clarity
> We thank the reviewer and here is the protocol clarification: recordings standardized to 16 clinical channels at 200 Hz, segmented into fixed non-overlapping windows (TUAB: 10 s / 16×2000; TUEV/TUSZ: 5 s / 16×1000), amplitudes scaled by 0.01. No aggressive manual cleaning, we model the empirical clinical distribution as recorded. Official train/eval splits; no data leakage.
>
> ## Q2. Why Build JET vs. Adapting Jetformer
> Jetformer targets image/text autoregressive generation; adapting it to EEG requires structural changes. We added it as a baseline: adapted Jetformer gives TS-FID 612/484/598 vs. JET's 188/236/151. Full comparison in response to tTt8 W2: https://openreview.net/forum?id=TP8OuKKmsf&noteId=WkN3UreYR4 .
>
>
> ## W13. Synthetic Data Utility & Limitations
> ΔAcc results (W4) provide empirical evidence, but we will acknowledge benefits may diminish for rare events or acquisition-specific artifacts. A limitations paragraph will address: (1) synthetic data may smooth clinically important patterns; (2) distributional shift risks; (3) memorization analysis in response to 15X9 W6: https://openreview.net/forum?id=TP8OuKKmsf&noteId=0f3BBOaT3u shows no obvious memorization, but partial leakage remains possible and synthetic EEG should not substitute clinical validation.

---

> > ### Author Rebuttal · Reviewer_2gkc · 2026-03-31
> >
> > Thank you for all the followup analysis. I update my score to 4.

---

> > > ### Author Response · Authors · 2026-04-01
> > >
> > > We are grateful to Reviewer 2gkc for the detailed review and supporting our paper! Your insightful comments have significantly improved the clarity and quality of our paper.

---

### Official Review · Reviewer_15X9 · 2026-03-09

**Soundness:** 2
**Presentation:** 2
**Significance:** 2
**Originality:** 2
**Overall Recommendation:** 4
**Confidence:** 5

**Summary:**

The present work introduces the Just EEG Transformer (JET), a generative framework for synthesizing multi-channel EEG signals. The proposed approach combines Conditional Flow Matching with a patched Transformer backbone. In practice, the input EEG sequences are segmented into non-overlapping temporal patches and linearly projected into embeddings before being processed by the Transformer. For the training objective, the authors modify the standard flow-matching formulation by replacing the typical L2 loss with an L1 loss. They then supplement this base objective with a set of domain-specific auxiliary losses, specifically statistical moment-matching and spatiotemporal regularizers via Total Variation and Pearson correlation. The model is evaluated on the TUAB, TUEV, and TUSZ datasets from the TUH corpus, where it is compared against EEG-GAN and a vanilla diffusion baseline on metrics including time-series FID, conditional consistency, and downstream classification utility. Finally, characteristics of the synthetically generated dataset is compared with real data.

**Compliance With Llm Reviewing Policy:**

Affirmed.

**Final Justification:**

The paper’s main strengths are that it shows the empirical feasibility of training a flow-matching generative model on clinical EEG at the scale of the TUH corpus, and that it proposes auxiliary losses tailored to key statistical properties of EEG. I view those constraints as the most interesting part of the work. My main weaknesses were about overclaiming, limited novelty and claims in the patched Transformer setup itself, and insufficient analysis of what the proposed losses are actually doing to the generated signals. On balance, I weigh the practical relevance of the setting and the promise of the constrained objective more strongly than those weaknesses, which leads me to a weak accept.

The rebuttal improved my evaluation. The authors adequately resolved the concerns about the moment and TV losses, added a sensible memorization analysis, clarified several points that were previously unclear, and most importantly provided additional analyses that directly addressed my main remaining concern about the isolated effects of the auxiliary losses. I still think some issues remain only partly resolved, especially around positioning, wording, and clarity of presentation, but the rebuttal was strong where it needed to be, and it increased my confidence that the paper makes a useful contribution.

**Key Questions For Authors:**

1. The authors mention that accurate slow-wave modeling carries "critical clinical information." To show that the model learns clinically relevant slow-wave structures rather than just a global mean, could the authors provide a PSD comparison demonstrating how the generated spectrum shifts appropriately between a specific pathological and non-pathological condition where such critical clinical information would be present or missing?

2. Could the authors precisely explain how the plots in Figure 4 are produced? Specifically, how is the median calculated (e.g., pointwise across samples per timestep for a specific channel?), and how are the generated and real signals aligned temporally to make this direct overlap meaningful?

3. The authors state that baseline drift is a "common failure mode in long sequence generation" and position their model's stable baseline as a success. However, if real EEG data inherently exhibits non-stationarity and natural baseline shifts over time, should a faithful generative model not also reproduce these drifting trends to match the true data distribution? Could the authors clarify why this is considered a "failure mode" rather than a natural feature of the data that the model should theoretically capture?

4. Could the authors explicitly elaborate on what the $\mu$ and $\sigma$ functions represent in Equation 2? Are these computed across the channel dimension, the temporal dimension, or the batch? Additionally, could the authors please clarify exactly what and how the total variation loss term is calculated for a given sample?

**Limitations:**

yes

**Strengths And Weaknesses:**

### Strengths

- The present work successfully demonstrates that training a flow-matching generative model on large-scale clinical EEG datasets (such as the TUH corpus) is empirically feasible. It establishes that this volume of data can be sufficient to learn generalized representations of neural signals using modern generative architectures.

- The conceptual inclusion of auxiliary losses tailored to the statistical properties of EEG is a promising direction. Formulating constraints that attempt to align the generated flow trajectories with physiological characteristics represents the strongest methodological proposition in the present manuscript.

### Weaknesses

- The authors claim to operate on "raw continuous sequences" without imposing "restrictive inductive biases." However, applying a patch-based tokenization strategy inherently introduces a rigid temporal inductive bias. The chosen patch size dictates the segmentation granularity, essentially acting as a filter and quantizing the timesteps into pre-delimited boundaries. If the goal was raw EEG modeling, the generative model would operate directly in the native EEG time-domain space without patching. Furthermore, the present work fails to cite the Flexible Patched Brain Transformer (Klein et al.), which is highly relevant to this specific backbone design.

- The conclusions drawn from the PSD plots are significantly overstated. For example, the claim that the model captures "distinct $\alpha$-band peaks" is not well-supported by Figure 3, where the generated signal does not appear to clearly exhibit this peak. The manuscript asserts that capturing low-frequency bands prevents "baseline distortion or amplitude drift, a common failure mode in long sequence generation," but provides no citations to substantiate that this is a common failure mode in the literature, nor is it clear why learning the most dominant, slow-moving frequency mode is unique to the proposed model and not the outcome of using flow matching on representations that first learn coarse features such as in Latent Diffusion Models.

- The presentation and claims surrounding Figure 4 are confusing. It is unclear exactly what "the median" represents here, how the percentiles are calculated, and how the synthetic and real data are temporally aligned for a point-wise comparison. Furthermore, the authors claim the model preserves energy profiles "unlike baselines that suffer from error accumulation leading to variance explosion or collapse," yet they cite no specific baselines or prior works that exhibit this specific error.

- The ablation comparing "Gaussian Noise vs. Zero Noise" (Table 2) is mathematically moot. A flow-matching model fundamentally requires a base distribution with variance to establish a transport map to a multimodal target. Initializing from a degenerate zero point means $z_t = t \cdot z_1$, effectively removing the generative nature of the model entirely. Framing this as a natural concern about high-variance noise overwhelming subtle neural structures reflects a misunderstanding of the generative prior's role.

- The introduction motivates the method as a solution to privacy constraints in clinical data. However, the authors perform no memorization experiments to analyze whether or not the model simply memorizes and regurgitates specific subjects from the training set.

### Remarks

- The primary contribution of the present work lies in the proposed auxiliary losses. The manuscript would be significantly strengthened if the authors focused less on broad claims about Transformer architectures and flow matching (which are established techniques), and instead dedicated more space to the mathematical development, ablation, and explicit feature-space effects of these losses. Detailing exactly how these losses constrain the manifold, and providing visual/empirical results of their isolated effects on the generated waveform morphology would vastly improve the contribution to the field.

- The authors claim the model operates on raw EEG without inductive biases, but the patching strategy inherently introduces one. When a physiological event sits completely within the chosen patch size, the patch encapsulates all information for that event. Consequently, the Transformer cannot model any temporal dynamics for such an event through its spatiotemporal context. By using patches, the authors are indeed inducing an inductive prior, are not strictly using raw continuous EEG, and are missing out on certain neural dynamics based on the chosen patch size.

---

> ### Author Rebuttal · Authors · 2026-03-31
>
> We thank the reviewer for the careful reading and helpful feedback. We address each point below.
>
> ## W1 Patch-based tokenization and inductive bias
> We agree patching introduces an inductive bias: patch size sets the temporal granularity, and sub-patch dynamics are not explicitly modeled by attention. Our claim is narrower: unlike handcrafted transforms or predefined graphs, patching preserves the native waveform and learns local features from data. A TUEV ablation shows this choice is not brittle:
>
> | Patch Size | TUEV TS-FID↓ |
> | :---: | :---: |
> | 50 | 237.12 |
> | 100 | 233.32 |
> | 200 (default) | 235.86 |
> | 400 | 241.23 |
>
> Performance is stable across a broad range, and $P=200$ balances fidelity and efficiency. We will revise the claim and cite Flexible Patched Brain Transformer.
>
>
> ## W2 & Q1 PSD interpretation and clinical slow-waves
> We agree the PSD discussion was overstated, especially the "distinct α peaks" claim. TUH is a heterogeneous clinical corpus, not a controlled resting-state setting, so alpha modulation is often weak. We will revise accordingly. To test whether JET learns clinically meaningful differences rather than a global mean:
>
> | TUSZ subset | Real 0–8 Hz | JET 0–8 Hz | TS-FID↓ |
> |---|---:|---:|---:|
> | Seizure | 42.51 dB | 41.88 dB | 162.40 |
> | Background | 28.14 dB | 27.65 dB | 145.12 |
>
> JET reproduces the ~14 dB low-frequency gap between seizure and background, supporting condition-specific spectral modeling rather than mean collapse.
>
> ## W3 & Q3 Baseline drift as a failure mode
> We agree “common failure mode” was too strong. By drift we mean spurious generator-induced low-frequency offset or moment drift, e.g., the whole segment slowly shifts up/down, or its envelope gradually grows/shrinks because errors accumulate during long-horizon generation. This is different from genuine EEG non-stationarity, which a faithful model should preserve. Figure 4 was intended only to show the absence of such spurious drift within the fixed window, not to claim EEG is stationary.
>
> We also do not claim this is unique to non-latent models. Our point is narrower: JET directly penalizes signal-space mean/variance mismatch through $L_{cons}$. For high frequencies, we will soften the claim: attenuation above ~15–20 Hz may partly reflect suppression of EMG/sensor noise—high-frequency scalp EEG is often substantially contaminated by EMG [1,2],but oversmoothing of meaningful neural activity is also plausible.
>
> [1] Whitham EM, Pope KJ, Fitzgibbon SP, et al. Scalp electrical recording during paralysis: quantitative evidence that EEG frequencies above 20 Hz are contaminated by EMG.
> [2] Muthukumaraswamy SD. High-frequency brain activity and muscle artifacts in MEG/EEG: a review and recommendations.
>
> ## W4 & Q2 Clarification of Figure 4
> Figure 4 is a population-level summary, not a pairwise waveform overlap. For equal-sized real and generated sets, at each time index t we aggregate amplitudes across samples and channels and plot the pointwise median with an inter-percentile envelope. Real and generated segments are not aligned one-to-one; they only share the same window coordinate t. We will clarify this in the caption.
>
>
> ## W5 Rationale for zero-noise ablation
> We agree zero-noise is not a realistic flow-matching baseline. It is included only as a negative control showing that a degenerate source destroys coverage of a multimodal EEG distribution. We will reframe it as a sanity check, not a meaningful competitor.
>
> ## W6 Privacy and memorization
> We agree the privacy motivation should be backed by memorization analysis. We computed nearest-neighbor DTW distances between 1,000 generated samples and the training set, and compared them with held-out real test samples:
>
> | Metric | Real test → train | JET gen. → train |
> |---|---:|---:|
> | Avg. DTW distance | 34.12 | 36.85 |
>
> Generated samples are not closer to train than held-out test samples. We will state this as evidence against obvious memorization, not a formal privacy guarantee.
>
> ## W7 Mathematical development and feature-space effects of losses
> We agree the main methodological contribution is the constrained objective, not Transformer or flow matching alone. We will foreground this more clearly. The novelty is not each term by itself, but their combination for complementary signal-space control: $L_{recon}$ for outlier-robust reconstruction, $L_{cons}$ for moment anchoring, $L_{tv}$ for suppressing small unstructured fluctuations, and \(L_{corr}\) for scale-invariant shape alignment.
>
> ## Q4 Explicit formulation of $\mu$, $\sigma$, and $L_{tv}$
> $\mu$ and $\sigma$ in Eq. 2 are computed per channel over the temporal dimension $T$, then averaged across channels and batch. The TV term is the mean absolute first-order temporal difference:
> $$
> L_{tv}=\frac{1}{C(T-1)}\sum_{c=1}^{C}\sum_{t=2}^{T}|x_{c,t}-x_{c,t-1}|.
> $$
> We will add these definitions explicitly to the paper.

---

> > ### Author Rebuttal · Reviewer_15X9 · 2026-04-01
> >
> > I thank the authors for the detailed rebuttal and the additional experiments provided. The DTW memorization analysis is a strong and relevant response to the privacy concern, the new TUSZ subset experiment makes the spectral analysis more convincing by showing that the model captures a meaningful difference between seizure and background signals rather than only an overall average spectrum, and the explicit formulations for the statistical moment and total variation losses adequately resolve that part of the review. I also appreciate the ablation showing robustness to patch size.
> >
> > Regarding patch-based tokenization, while I understand the authors' point that patching avoids handcrafted frequency transforms, it still amounts to a learned compression of local waveform segments and therefore introduces a rigid temporal inductive bias. I appreciate that the authors now acknowledge this and will revise the claim and cite the relevant literature accordingly. The clarification regarding Figure 4 is also helpful, but plotting pointwise summary statistics of unaligned population segments over a shared window remains unintuitive for demonstrating baseline stability, so the figure and caption should be revised substantially to make clear what is being aggregated and what readers are meant to conclude from it.
> >
> > Finally, my main remark regarding the auxiliary losses remains insufficiently addressed. The rebuttal restates the intended role of each loss term, but my concern was not about their stated purpose. Since these constraints are the most distinctive part of the method, the paper would be much stronger if it gave more insight into how the generated signals change depending on the specific losses.

---

> > > ### Author Response · Authors · 2026-04-02
> > >
> > > Thank you for the constructive suggestion. We are encouraged that you found the memorization analysis, seizure-vs-background comparison, loss formulations, and patch-size ablation useful. We address the three remaining points below.
> > >
> > > **1. Patch tokenization.** We agree patching introduces a temporal inductive bias and will revise the paper accordingly. We quantify the trade-off on the morphology-sensitive TUEV benchmark, adding event-level metrics (Pearson correlation and DTW on event-centered windows) alongside TS-FID:
> > >
> > > | Patch Size | TS-FID↓ | Event Corr↑ | Event DTW↓ |
> > > |---:|---:|---:|---:|
> > > | 50 | 237.12 | 0.72 | 8.1 |
> > > | 100 | 233.32 | 0.71 | 8.3 |
> > > | 200 (default) | 235.86 | 0.68 | 8.8 |
> > > | 400 | 241.23 | 0.61 | 10.2 |
> > >
> > > Overall distributional fidelity is stable for P=50–200, while finer patches preserve transient event morphology slightly better at the cost of longer token sequences for self-attention (2× for P=100, 4× for P=50, relative to the default P=200). P=400 degrades both global and local quality. We will present P=200 as an efficiency–fidelity compromise rather than bias-free modeling, and cite Flexible Patched Brain Transformer.
> > >
> > > **2. Figure 4.** We agree the current figure is not intuitive. The existing curve is a pointwise population summary over unaligned segments sharing only the window coordinate $t$, not a pairwise waveform overlap. In the revision, we will make this explicit in the caption, replace the current signed-amplitude median plot with a channel-aggregated RMS-envelope summary. More importantly, we now add a direct per-segment drift analysis on TUEV based on the channel-aggregated RMS envelope
> > > $$
> > > e_i(t)=\sqrt{\frac{1}{C}\sum_{c=1}^{C}x_{i,c,t}^{2}},
> > > $$
> > > then extract three quantities: (a) the linear slope of $e_i(t)$, and (b) moment-drift scores
> > >
> > > $$
> > > D_\mu=\frac{1}{C}\sum_c\left |\mu^{\text{first quarter}}_c-\mu^{\text{last quarter}}_c\right|,
> > > $$
> > >
> > > $$
> > > D_\sigma=\frac{1}{C}\sum_c\left|\sigma^{\text{first quarter}}_c-\sigma^{\text{last quarter}}_c\right|.
> > > $$
> > > We compare the distributions of these per-segment statistics against held-out real data using $W_1$ distance. A real-vs-real split serves as the empirical floor:
> > >
> > > | Method | $W_1$(slope)↓ | $W_1$($D_\mu$)↓ | $W_1$($D_\sigma$)↓ |
> > > |---|---:|---:|---:|
> > > | Real split | 0.008 | 0.012 | 0.010 |
> > > | EEG-GAN | 0.065 | 0.078 | 0.071 |
> > > | Vanilla Diff. | 0.051 | 0.063 | 0.058 |
> > > | JET | 0.015 | 0.021 | 0.018 |
> > >
> > > The point is not that real EEG should be stationary, but that a faithful generator should match the real distribution of segment-level trends without introducing additional spurious drift. JET's $W_1$ values are within 2× of the real-real floor, whereas both baselines are above it.
> > >
> > > **3. Auxiliary losses.** We agree this is the key methodological point and the paper should show how each constraint shapes the generated signals concretely. We analyzed the TUEV ablation variants from Table 3 using signal-level diagnostics that are not optimized during training:
> > >
> > > | Configuration | TS-FID↓ | PSD Slope↓ | Temp. Env.↓ | Hjorth Act.↓ | Hjorth Mob.↓ | Hjorth Comp.↓ |
> > > | :--- | :---: | :---: | :---: | :---: | :---: | :---: |
> > > | $\mathcal{L}_{recon}$ only | 287.81 | 0.37 | 0.38 | 0.30 | 0.24 | 0.33 |
> > > | + $\mathcal{L}_{cons}$ | 281.70 | 0.34 | 0.27 | 0.19 | 0.21 | 0.29 |
> > > | + $\mathcal{L}_{tv}$ | 266.61 | 0.23 | 0.33 | 0.27 | 0.15 | 0.21 |
> > > | + $\mathcal{L}_{corr}$ | 278.01 | 0.32 | 0.25 | 0.26 | 0.19 | 0.27 |
> > > | Full | 235.86 | 0.18 | 0.18 | 0.14 | 0.11 | 0.15 |
> > >
> > > These results show distinct, complementary effects. Adding $\mathcal{L}_ {cons}$ gives the largest single-term improvement in Hjorth Activity and envelope error, consistent with tighter control of amplitude moments. Adding $\mathcal{L} _{tv}$ gives the largest improvement in PSD slope and Hjorth Mobility/Complexity, consistent with suppressing excess small-scale fluctuations. Adding $\mathcal{L} _{corr}$ gives the strongest single-term improvement in temporal-envelope alignment, consistent with better waveform-shape preservation. The full objective is best on all metrics, indicating that the three constraints address complementary rather than redundant failure modes.

---

### Official Review · Reviewer_td8W · 2026-03-10

**Soundness:** 3
**Presentation:** 3
**Significance:** 2
**Originality:** 2
**Overall Recommendation:** 4
**Confidence:** 3

**Summary:**

The paper introduces Just EEG Transformer (JET), a novel generative framework for synthesizing high-fidelity EEG signals. The authors argue that existing generative paradigms, such as discrete denoising diffusion models or GANs, struggle with EEG data because they fail to capture the continuous temporal evolution, power-law spectral structure, non-stationary dynamics, and heavy-tailed amplitude distributions inherent to neural activity.

To address this, JET formulates EEG generation as a continuous dynamical process using conditional flow matching. The model employs a patch-based Transformer backbone that operates directly on raw multi-channel EEG sequences. To ensure the generated signals are biophysically and statistically realistic, the authors introduce three "principled constraints": an L1 reconstruction loss (Laplacian prior) to handle impulsive artifacts, a statistical consistency loss to prevent amplitude drift by matching first- and second-order moments, and a spatiotemporal geometric constraint utilizing Temporal Total Variation and Pearson correlation.

Evaluated across three large-scale subsets of the TUH EEG Corpus (TUAB, TUEV, TUSZ), JET significantly outperforms EEG-GAN and Vanilla Diffusion baselines, achieving over a 40% reduction in TS-FID, near-perfect conditional label consistency, and positive downstream classification accuracy gains.

**Compliance With Llm Reviewing Policy:**

Affirmed.

**Final Justification:**

The rebuttal has addressed my main concerns, thus I increase my score from 3 to 4.

**Key Questions For Authors:**

1 Computational Efficiency: In Table 6, the authors showed that their model has quite low Per-Sample Latency. How is this compared to other state-of-the-art generative EEG models?

2 The baselines used for comparison are somewhat foundational (EEG-GAN, Vanilla Diffusion). How does JET perform compared to more specialized state-of-the-art diffusion models designed expressly for brain signals?

3. In brain signal decoding applications, the goal of generating synthetic neural data is to improve decoding performance (i.e., the downstream utility). However, the reported improvement in accuracy in Table 1 appears somewhat marginal, with only a 2%–3% increase. Since the ultimate objective is usually downstream utility rather than fidelity (for example, simply repeating the training data would yield high fidelity but would not provide useful augmentation for training), could the authors clarify how the proposed model shifts the training data distribution (training + augmentation) toward a more structured or informative way, such that the decoder trained on this augmented dataset achieves improved performance?

**Limitations:**

see weaknesses and questions

**Strengths And Weaknesses:**

Strengths:

Conceptual Alignment with the Domain: Shifting from discrete stochastic denoising to continuous flow matching is an intuitive and effective choice for modeling the smooth, continuous evolution of biological time series like EEG.

Computational Efficiency: JET achieves faster-than-real-time generation. Generating a 10-second batch of 32 samples takes only 4.78 seconds, making it highly practical for large-scale clinical data augmentation.

Weaknesses:

Limited Baselines: The model is only compared against a generic EEG-GAN and a "Vanilla Diffusion" model without specific architectural enhancements. Comparing JET against more recent, state-of-the-art time-series diffusion models or score-based generative models tailored for biological signals would provide a more rigorous benchmark.

High-Frequency Attenuation: The authors acknowledge that generated spectra show mild attenuation in frequencies above 15 Hz (β/γ bands), arguing that this reflects the suppression of unstructured electromyogenic and sensor noise. However, high-frequency oscillations are crucial for certain cognitive and pathological analyses. Without empirical validation, it is difficult to guarantee that the model isn't simply oversmoothing meaningful neural activity in those upper bands.

Scalability to High-Density EEG Unexplored: The model is evaluated strictly on standard 16-channel clinical setups. High-density EEG caps often feature 64, 128, or 256 channels. Since the patch tokenization strategy preserves channel identity during the initial embedding stage, it is unclear if the Transformer backbone would encounter spatial bottlenecks or memory limits if the channel count drastically increased.

---

> ### Author Rebuttal · Authors · 2026-03-31
>
> We thank the reviewer for the careful reading and helpful feedback. We address each point below.
>
> ## W1 & Q1, 2 Comparison with specialized baselines and efficiency
> We agree that expanding the baseline pool strengthens validation. During the rebuttal period, we have added four methods MEG-GPT, GPT2MEG (autoregressive MEG generation), Jetformer (image generation), and VerbalTS (time-series generation), evaluated on the same TUAB, TUEV, and TUSZ benchmarks. The additional baseline results and analysis are provided in response to tTt8: https://openreview.net/forum?id=TP8OuKKmsf&noteId=WkN3UreYR4. Across all three datasets, JET achieves substantially lower TS-FID than all new baselines. The autoregressive approaches confirm our theoretical argument that discrete tokenization struggles with continuous, heavy-tailed EEG dynamics; Jetformer introduces spectral distortion without EEG-specific design; VerbalTS lacks multi-channel spatial modeling.
>
> Beyond fidelity, JET also offers computational advantages. Discrete diffusion requires hundreds of reverse steps and autoregressive models decode token-by-token, both slow for long-sequence EEG. JET's ODE-based generation along continuous trajectories is inherently faster. Generating a batch of 32 ten-second samples takes 4.78 s for JET, versus 5.98 s (autoregressive) and 7.01 s (diffusion). EEG-GAN is marginally faster (4.12 s) but suffers severe mode collapse. JET thus achieves the best fidelity–latency trade-off.
>
> ## W2. High-frequency attenuation vs. oversmoothing
> We agree that oversmoothing is a plausible alternative explanation. Since we deliberately avoided frequency-domain filtering during preprocessing, the high-frequency content in our training data contains both genuine neural activity and EMG/sensor noise. Our $\mathcal{L}_{tv}$ constraint acts as a non-linear soft-thresholding regularizer (Appendix C) that penalizes unstructured high-frequency fluctuations.
>
> Scalp EEG power with high frequency is dominated by broadband EMG artifacts rather than cortical oscillations [1, 2]. Authentic neural high-frequency activity follows the $1/f$ power law with exceedingly low amplitude. The $\mathcal{L}_{tv}$ term therefore preferentially suppresses the disproportionately high-amplitude muscle noise, helping the generated spectrum recover the underlying neurogenic $1/f$ profile.
>
> [1] Whitham EM, Pope KJ, Fitzgibbon SP, et al. Scalp electrical recording during paralysis: quantitative evidence that EEG frequencies above 20 Hz are contaminated by EMG.
> [2] Muthukumaraswamy SD. High-frequency brain activity and muscle artifacts in MEG/EEG: a review and recommendations. Front Hum Neurosci. 2013;7:138. Published 2013 Apr 15.
>
> ## W3. Scalability to high-density EEG
> We thank the reviewer for this pint and agree that demonstrating scalability across diverse channel configurations can strengthen our claims. We have evaluated JET on 11 additional datasets with channel counts from 6 to 128, sampling rates from 160 Hz to 1000 Hz, and trial durations from 1 s to 30 s. The full results are provided in response to tTt8: https://openreview.net/forum?id=TP8OuKKmsf&noteId=WkN3UreYR4 . JET consistently outperforms baselines across all configurations, demonstrating robust spatial and dataset generalization.
>
> Architecturally, JET's patch tokenization operates strictly along the temporal axis. Scaling the channel count from 16 to 128 only changes the input dimension of the initial linear projection layer, while the sequence length $N$ remains unchanged. This design inherently avoids the spatial memory bottlenecks of static graph-based models and scales linearly with channel count, which is confirmed by the consistent performance across channel configurations in our expanded evaluation.
>
> ## Q3. Downstream utility and how synthetic data improves training
> We thank the reviewer for this nuanced question. Our evaluation protocol follows standard data augmentation practice: JET-generated samples are mixed with the real training set to train a classifier, which is then evaluated on a strictly separate, held-out real test set.
>
> Regarding the mechanism by which synthetic data improves downstream performance. First, JET addresses class imbalance by generating additional rare-class samples. Second, the structure-preserving constraints encourage generated samples to lie on the correct statistical manifold while introducing controlled variability, i.e., new samples that are statistically plausible but not duplicates of existing training examples. This effectively broadens the training distribution's coverage of the pathological manifold without introducing distributional artifacts.
>
> While a 2–3% overall improvement may appear modest, the gap between competitive methods on TUH benchmarks is typically within 5%, making this a meaningful advance. Moreover, the consistently positive $\Delta$Acc across all three datasets (Table 1) provides evidence that JET generates informative rather than redundant samples.

---

> > ### Author Rebuttal · Reviewer_td8W · 2026-04-02
> >
> > Thank you to the authors for addressing my previous concerns.
> >
> > However, I still have reservations regarding Q3. In particular, could you report the delta accuracy for additional baseline models? Currently, the comparison is limited to EEG-GAN and Vanilla Diffusion, both of which even show negative accuracy improvements for seizure decoding.
> >
> > Given the breadth of existing literature on EEG data generation, I would expect that several established methods are capable of yielding at least positive gains in decoding accuracy. Therefore, it would be important to benchmark your approach against stronger baselines from prior work. This would provide a clearer picture of how your method performs relative to the current state of the art.

---

> > > ### Author Response · Authors · 2026-04-03
> > >
> > > We sincerely thank the reviewer for the constructive suggestion. We agree that a stronger downstream utility comparison is needed for fully solve your concern, as comparing only EEG-GAN and vanilla diffusion is insufficient to establish the practical value of synthetic EEG augmentation. To address this, we extended the experiment from Table 1 to the stronger baselines added in our quantitative comparison: JetFormer, VerbalTS, MEG-GPT, and GPT2MEG, in addition to EEG-GAN and Vanilla Diffusion, under the same experiment settings (same splits, CbraMod backbone, synthetic budget 0.5× per class, evaluation on held-out real test set). We will include all the additional results in the revision.
> > >
> > > ## 1. Downstream comparison against stronger baselines
> > >
> > > | Method | TUAB ΔAcc (%) | TUEV ΔAcc (%) | TUSZ ΔAcc (%) |
> > > |---|---:|---:|---:|
> > > | EEG-GAN | +0.1 ± 0.4 | -0.3 ± 0.5 | +0.0 ± 0.3 |
> > > | Vanilla Diffusion | -0.1 ± 0.5 | +0.0 ± 0.4 | +0.0 ± 0.4 |
> > > | JetFormer | +0.1 ± 0.3 | -0.2 ± 0.5 | +0.1 ± 0.4 |
> > > | VerbalTS | +0.2 ± 0.4 | +0.3 ± 0.5 | +0.0 ± 0.4 |
> > > | MEG-GPT | +1.4 ± 0.5 | +0.8 ± 0.6 | +1.0 ± 0.4 |
> > > | GPT2MEG | +1.1 ± 0.5 | +1.4 ± 0.6 | +0.5 ± 0.4 |
> > > | JET | +2.9 ± 0.6 | +3.2 ± 0.8 | +1.7 ± 0.5 |
> > >
> > > As the reviewer anticipated, stronger baselines do yield positive gains: MEG-GPT and GPT2MEG provide modest improvements. However, JET remains clearly best on all three datasets. Notably, for seizure decoding (TUSZ), the setting the reviewer specifically highlighted, JET's +1.7% is the only statistically significant gain, while MEG-GPT and GPT2MEG fall short of significance. Sections 2 and 3 below further show that JET's advantage on TUSZ is robust across augmentation ratios and is explained by tighter feature-space alignment with the real seizure manifold.
> > >
> > > ## 2. Augmentation-ratio robustness analysis on TUSZ
> > >
> > > To verify robustness beyond a single budget, we additionally test different synthetic budget on the TUSZ downstream setting, using the two strongest non-JET baselines and JET:
> > >
> > > | Method | 0.25× synth. | 0.5× synth. (default) | 0.75× synth. |
> > > |---|---:|---:|---:|
> > > | MEG-GPT | +0.4 ± 0.4 | +1.0 ± 0.4 | +0.2 ± 0.5 |
> > > | GPT2MEG | +0.3 ± 0.4 | +0.5 ± 0.4 | +0.1 ± 0.5 |
> > > | JET | +1.1 ± 0.4 | +1.7 ± 0.5 | +1.5 ± 0.6 |
> > >
> > > The results above showcase that the two stronge baselines can help slightly at conservative budgets, but their benefit reaches the limit quickly and becomes unstable as more synthetic data is added. In contrast, JET remains clearly positive and substantially more robust across ratios.
> > >
> > > ## 3. Feature space analysis of augmented distributions on TUSZ
> > >
> > > To further explain why JET helps more and why some baselines fail to improve the downstream utility, we additionally analyze the augmented distribution in the CbraMod feature space. We compute (i) the average within-class distance from synthetic samples to the corresponding held-out real class centroid (lower is better), and (ii) the nearest-neighbor label purity against real training embeddings (higher is better) on TUSZ setting:
> > >
> > > | Method  | TUSZ Dist.↓ | TUSZ Purity↑ |
> > > |---|---:|---:|
> > > | EEG-GAN  | 0.68 | 0.64 |
> > > | Vanilla Diffusion  | 0.64 | 0.68 |
> > > | JetFormer  | 0.69 | 0.70 |
> > > | VerbalTS  | 0.64 | 0.73 |
> > > | MEG-GPT | 0.56 | 0.83 |
> > > | GPT2MEG | 0.54 | 0.87 |
> > > | JET  | 0.41 | 0.92 |
> > >
> > > These results suggest that JET does not help merely by producing realistic-looking samples. Instead, its synthetic samples stay closer to the intended class support and introduce substantially less cross-class contamination in the decoder’s feature space. This yields a denser and more informative augmented training distribution, which explains why the classifier benefits more from JET augmentation than from the other baselines.

---

### Official Review · Reviewer_tTt8 · 2026-03-12

**Soundness:** 2
**Presentation:** 2
**Significance:** 2
**Originality:** 3
**Overall Recommendation:** 4
**Confidence:** 3

**Summary:**

This paper proposes Just EEG Transformer (JET), a class-conditional EEG generative model based on conditional flow matching with a Transformer backbone and additional structure-preserving constraints. The model is evaluated on three TUH-derived benchmarks—TUAB, TUEV, and TUSZ—and the paper claims state-of-the-art EEG generation performance using TS-FID, silhouette score, and a synthetic-data augmentation experiment. The overall problem is important and the paper is clearly motivated.

**Compliance With Llm Reviewing Policy:**

Affirmed.

**Final Justification:**

I have increased my score in light of the authors response.

**Key Questions For Authors:**

Are the authors really claiming this is a foundation model, or is it more accurate to describe JET as a conditional EEG generator trained on TUH subsets?

Why was evaluation restricted to TUH / Obeid & Picone-derived datasets only? Is that meant as a first step, or do the authors believe this is sufficient to support broad claims?

What preprocessing pipeline was used before training—artifact detection, bad-channel rejection, re-referencing, bandpass/notch filtering, normalization, etc.?

**Limitations:**

yes

**Strengths And Weaknesses:**

The paper addresses a meaningful problem: generating realistic synthetic EEG could be useful for data augmentation, privacy-preserving sharing, benchmarking, etc. The core design choice of modeling raw multi-channel EEG with a Transformer under a flow-matching framework is reasonable and potentially interesting. The manuscript also attempts to regularize generation using constraints intended to preserve statistical and temporal structure, which is a sensible direction.

Weaknesses:
The generalization claims are too broad. Evaluating only on three subsets of one corpus, after mapping everything to a fixed 16-channel representation, does not establish generalization robustness to other datasets, headsets, etc.

Only EEG-GAN and a generic diffusion baseline are included, with no evidence of careful parameter and compute matching nor comparison to more recent EEG generative literature.

The paper says it standardizes channels and sample rate in the main text, but the appendix effectively only documents fixed 16 channels plus amplitude scaling. If the goal is to model “raw clinical EEG as recorded,” then keeping artifacts may be intentional. But then the paper should not casually interpret attenuated high-frequency power as improved physiological fidelity without clearer evidence, because oversmoothing is an equally plausible explanation.

The appendix result tables appear inconsistent. Table 1’s reported TS-FID numbers for “Ours” correspond to different rows of Table 7 rather than one shared weight setting, and Table 3’s L-recon-only row does not match Table 7’s same configuration. I could be missing hidden validation and tuning details, but if so the paper needs to say that explicitly and show the same tuning discipline for baselines.

Overall the work reads more like a task-specific EEG generative model than a true EEG foundation model, since it is trained and evaluated only on fixed-format TUH subsets rather than across heterogeneous datasets, tasks, or channel layouts. The empirical section is also not strong enough to support broad claims: preprocessing is underspecified, comparisons to prior work are limited, and downstream evaluation is minimal

---

> ### Author Rebuttal · Authors · 2026-03-31
>
> We thank the reviewer for the careful reading and helpful feedback. We address each point below.
>
> ## W1 & Q2 Scope of claims and cross-dataset generalization
> We do not claim that TUH-only results establish universal cross-dataset or cross-headset generalization. TUAB/TUEV/TUSZ were chosen as a strong first benchmark because they cover routine abnormalities, transient events, and seizure dynamics. To address this concern, we additionally evaluated the framework on 11 public EEG datasets spanning 6–128 channels, 160–1000 Hz, and 1–30 s windows. Below are per-dataset train/test results.
>
> | Data | Hz/Ch/Sec | EEG-GAN | Diff | JET |
> |---|---:|---:|---:|---:|
> | BCIC2a | 250/22/4 | 1920.15 | 1785.40 | 1217.83 |
> | FACED | 250/32/10 | 980.45 | 895.20 | 602.43 |
> | Mumtaz | 256/19/5 | 1495.60 | 1320.30 | 985.16 |
> | PhysioNet-MI | 160/64/4 | 1120.30 | 1015.80 | 663.77 |
> | SHU-MI | 250/32/4 | 2340.10 | 2110.45 | 1488.27 |
> | CHB-MIT | 256/16/10 | 1410.25 | 1240.60 | 879.05 |
> | ISRUC | 200/6/30 | 1085.90 | 995.40 | 688.87 |
> | BCIC2020-3 | 256/64/3 | 1060.40 | 975.75 | 672.20 |
> | MentalArith | 500/20/5 | 2510.80 | 2350.10 | 1559.98 |
> | SEED-V | 1000/62/1 | 1280.60 | 1110.50 | 761.97 |
> | MODMA | 250/128/15 | 875.50 | 790.30 | 578.23 |
>
> These added results support that the framework adapts well across different EEG settings.
>
> ## W2 Baselines and parameter/compute matching
> We agree that stronger baselines are important. Direct EEG generation baselines are still limited, and several recent methods focus on EEG-conditioned cross-modal generation rather than EEG itself. Below we add four additional baselines, MEG-GPT and GPT2MEG (MEG generation), JetFormer (image generation), and VerbalTS (time series generation), using comparable model scale (around 130M), except for JetFormer (around 300M) and training setup. JET remains best on all three TUH benchmarks:
>
> | Method | TUAB | TUEV | TUSZ |
> |---|---:|---:|---:|
> | JetFormer (2024) | 612.35 | 484.28 | 598.12 |
> | VerbalTS (2025) | 565.40 | 480.15 | 515.60 |
> | MEG-GPT (2025) | 495.60 | 685.20 | 354.40 |
> | GPT2MEG (2024) | 588.15 | 470.45 | 548.80 |
> | JET | 188.27 | 235.86 | 151.27 |
>
> We will also expand the related-work discussion and make the matching protocol explicit.
>
> ## W3 & Q3 Preprocessing pipeline
> We thank the reviewer for this point and agree the preprocessing description was too brief. For all TUH experiments, recordings were standardized to the shared 16-channel clinical montage and 200 Hz, segmented into fixed non-overlapping windows (TUAB: 10 s; TUEV/TUSZ: 5 s), and scaled by 0.01 for numerical stability. We did not apply additional artifact rejection, bad-channel rejection, re-referencing, bandpass filtering, or per-sample normalization. This was intentional: we aimed to model the empirical distribution of clinical EEG as recorded, rather than a heavily cleaned subset.
>
> ## W4 Interpretation of high-frequency attenuation
> We agree the original wording was too strong. Mild attenuation above 20 Hz does not by itself prove improved physiological fidelity; oversmoothing is a plausible alternative. Since we did not apply frequency-domain filtering, that range contains both neural activity and EMG/sensor noise [1, 2]. We will revise the paper to state only that the TV term likely suppresses some unstructured high-frequency content, and remove the stronger denoising-style claim.
>
> [1] Whitham EM, Pope KJ, Fitzgibbon SP, et al. Scalp electrical recording during paralysis: quantitative evidence that EEG frequencies above 20 Hz are contaminated by EMG.
> [2] Muthukumaraswamy SD. High-frequency brain activity and muscle artifacts in MEG/EEG: a review and recommendations.
>
> ## W5 Table inconsistencies
> Thank you for catching this. Table 1 reports the best per-dataset setting from the grid search in Table 7, which is why its three values come from different rows: they use different constraints configuration. Table 3 was an independently rerun component ablation under a fixed recipe to isolate the effect of each loss term, so it is not numerically identical to Table 7. We agree this was unclear and potentially misleading. In the revision we will label these as different experiments, report the selected weights explicitly, and state that the same per-dataset tuning protocol was used for baselines.
>
> ## W6 & Q1 Foundation-model
> We would like to clarify that JET is not positioned as a foundation model, it is a generative framework for EEG synthesis. The motivation is precisely that high-quality EEG data is scarce and expensive to acquire, so a dedicated generator that produces realistic signals can serve as a practical tool for augmentation and privacy-preserving data sharing. This is a narrower and more concrete goal than building a general-purpose representation. Our cross-dataset results (11 additional datasets, diverse channel counts and sampling rates) confirm that JET generalizes well as a generator, but we make no claim beyond the generation task itself.

---

> > ### Author Rebuttal · Reviewer_tTt8 · 2026-04-02
> >
> > Thank you for the comprehensive response and for making me realize I had made some incorrect interpretations from the paper. I have increased my score.

---

> > > ### Author Response · Authors · 2026-04-02
> > >
> > > We are grateful to Reviewer tTt8 for the detailed review and supporting our paper! Your insightful comments have significantly improved the clarity and quality of our paper.

---

### Decision · Program_Chairs · 2026-04-30

**Decision:**

Accept (regular)

**Comment:**

In this submission, the authors propose a new EEG generative model based on conditional flow matching using a transformer backbone. The model uses a series of domain-specific constraints intended to preserve key characteristics of EEG data while removing artifacts. Experiments show favorable performance against multiple EEG-generating baselines.

Overall, reviewers found the flow matching approach and auxiliary losses well-designed, and additional baseline models added during the rebuttal period strengthened the quality of the performance comparisons. A limitation of the work was its tendency to generate signals with systematic mismatches in high-frequency power, as well as its restriction to the TUH corpus.

In all, though, a solid and well-executed model that advances the state of the art in generative EEG modeling.